# The second messenger signaling molecule cyclic di-AMP drives developmental cycle progression in *Chlamydia trachomatis*

**Junghoon Lee, Scot P Ouellette***

Department of Pathology, Microbiology, and Immunology, College of Medicine, University of Nebraska Medical Center, Omaha, United States

## eLife Assessment

In this **useful** study, ectopic expression and knockdown strategies were used to assess the effects of increasing and decreasing Cyclic di-AMP on the developmental cycle in Chlamydia. The authors **convincingly** demonstrate that overexpression of the dacA-ybbR operon results in increased production of c-di-AMP and early expression of the transitionary gene hctA and late gene omcB. Whilst the authors have attempted to revise the submission, the model proposed in the revised manuscript is still not fully supported by the data presented.

**\*For correspondence:**
scot.ouellette@unmc.edu

**Competing interest:** The authors declare that no competing interests exist.

**Abstract** The obligate intracellular bacterium *Chlamydia* alternates between two functional forms during its developmental cycle: elementary body (EB) and reticulate body (RB). However, the molecular mechanisms governing the transitions between these forms are unknown. Here, we present evidence that cyclic di-AMP (c-di-AMP) is a key factor in triggering the transition from RB to EB (i.e., secondary differentiation) in the chlamydial developmental cycle. By overexpressing or knocking down expression of c-di-AMP synthase genes, we made strains producing different levels of c-di-AMP, which we linked to changes in secondary differentiation status. Increases in c-di-AMP resulted in an earlier increase in transcription of EB-associated genes, and this was further manifested in earlier production of EBs. In contrast, when c-di-AMP levels were decreased, developmental cycle progression was delayed. Based on these data, we conclude there is a threshold level of c-di-AMP needed to trigger secondary differentiation in *Chlamydia*. This study identifies a mechanism by which secondary differentiation is initiated in *Chlamydia* and reveals a critical role for the second messenger signaling molecule c-di-AMP in this process.

## Introduction

*Chlamydia* species are major pathogens of humans and animals. These obligate intracellular bacteria share one key feature: their unique developmental cycle (see *Abdelrahman and Belland, 2005*). During this cycle, *Chlamydia* transitions between two different functional and morphological forms: the elementary body (EB), an infectious but non-dividing cell, and the reticulate body (RB), a dividing but non-infectious cell (*Figure 1A*). A third form, the intermediate body (IB) is a transitional form from the RB to EB. Besides these characteristics, EBs and RBs differ in other ways. For example, EBs are small (~0.3 μm), have a highly disulfide-crosslinked outer membrane (*Everett and Hatch, 1991*), and have DNA condensed by histone-like proteins (*Hackstadt et al., 1991*). In contrast, RBs are larger (~1 μm), have a Gram-negative cell envelope that lacks peptidoglycan (*Moulder, 1993*; *Fox et al., 1990*; *Barbour et al., 1982*), and have a dispersed chromosome. RBs divide by an asymmetric polarized division mechanism dependent on MreB-directed peptidoglycan synthesis specifically at the

**Figure 1.** Cyclic di-AMP accumulation is linked to secondary differentiation in *C. trachomatis*. (**A**) Defining characteristics of chlamydial elementary bodies (EBs) and reticulate bodies (RBs). (**B**) A hypothetical model representing the correlation between c-di-AMP levels and the timing of secondary differentiation. The dashed line represents a threshold level of c-di-AMP needed to drive secondary differentiation in a given RB. hpi = hours post-infection. (**C**) Measurement of c-di-AMP concentrations in uninfected (-Ctr) and infected (+Ctr) HeLa cell lysates. For infected HeLa cells, *C. trachomatis* serovar L2 (434/Bu) transformed with an mCherry-encoding construct was infected into HeLa cells, and expression of mCherry was induced at 10 hpi with 5 nM anhydrotetracycline (aTc). All samples were harvested at 16 and 24 hpi. (**D**) A schematic diagram of the constructs used in this study. All constructs used are aTc-inducible as shown by the $P_{tet}$ promoter. The location of the 6xH tag and the approximate location of the crRNA for the CRISPRi vectors are shown as well as the transmembrane domain of DacA. Diagram is not to scale. (**E**) Measurement of c-di-AMP concentrations in infected cell lysates from the strains shown in panel (**D**). *C. trachomatis* serovar L2 (434/Bu) transformed with the indicated constructs was infected into HeLa cells. At 10 hpi, expression of the construct was induced with 5 nM aTc, and the infected cells were harvested at 16 or 24 hpi. Levels of c-di-AMP in the supernatant were measured using ELISA. The left and right panels show the levels of c-di-AMP in the indicated strains at 16 and 24 hpi, respectively, on a log2 scale. For reference, $2^{10}=1024$, $2^{15}=32,768$, and $2^{20}=1,048,576$. The dashed line in both graphs represents the c-di-AMP level of the 24 hpi mCherry-expressing control that is associated with EB production. N=3. *p<0.05, **p<0.001, NS: Not significant via two-sample equal variance t-test compared to the mCherry control.

The online version of this article includes the following source data and figure supplement(s) for figure 1:

**Source data 1.** Source data for c-di-AMP experiments shown in *Figure 1*.

**Figure supplement 1.** The predicted transmembrane domains in DacA and YbbR.

septum (*Abdelrahman et al., 2016*; *Liechti et al., 2016*; *Ouellette et al., 2012*; *Lee et al., 2020*). Not surprisingly, *Chlamydia* expresses genes in a temporally defined manner that corresponds broadly with its developmental cycle (*Belland et al., 2003*; *Ouellette et al., 2006*). 'Early' genes (e.g., *euo*) are expressed immediately upon entry into a target host cell and are likely involved in establishing the intracellular niche of *Chlamydia*, the inclusion, and mediating primary differentiation from EB to RB. 'Mid' cycle genes (e.g., *mreB*, *clpPX*) facilitate RB replication and division and inclusion growth. 'Late' genes (e.g. *hctA*, *omcB*) are expressed when secondary differentiation is initiated to trigger EB formation. Although developmental gene expression has been characterized for decades, the signals or events that initiate differentiation from one form to the other are not known.

In a 2013 study, another defining characteristic of EBs and RBs was identified: their relative levels of the second messenger signaling molecule cyclic di-AMP (c-di-AMP) (see *Barker et al., 2013*). Barker et al. studied how IFNβ production is activated in cells infected with *C. trachomatis*. The authors

identified a role for the innate immune response protein, STING, which recognizes c-di-AMP. *Chlamydia* encodes a diadenylate cyclase enzyme, DacA, associated with c-di-AMP production, that had not been characterized. As part of their study, the authors determined that c-di-AMP accumulates over the course of the chlamydial developmental cycle, that EBs have high levels of this molecule whereas RBs have low levels, and that DacA is a diadenylate cyclase. It is unlikely that *Chlamydia* produces c-di-AMP only to signal host immune responses. Rather, the parsimonious interpretation is that *Chlamydia* uses this signaling molecule to regulate some aspect of its physiology and that activation of host signaling is 'accidental' – similar to activation of NOD2 by chlamydial peptidoglycan (*Girardin et al., 2003*; *Packiam et al., 2015*). However, no direct function of c-di-AMP in chlamydial biology has been defined.

Diverse functions of c-di-AMP have been reported in Gram-positive bacteria. For example, c-di-AMP is implicated in the response to changes in osmotic pressure. When the extracellular solute level is high, bacteria prevent dehydration by importing both extracellular solute molecules and cations (*Booth and Higgins, 1990*). In these mechanisms, c-di-AMP binds to proteins associated with K$^+$ uptake systems such as Ktr/Trk, KimA, Kup, and Kdp and inhibits their activities in *Bacillus subtilis*, *Staphylococcus aureus*, and *Lactococcus lactis* (*Kim et al., 2015*; *Gibhardt et al., 2019*; *Quintana et al., 2019*; *Moscoso et al., 2016*). Similarly, K$^+$ export mechanisms are also regulated by c-di-AMP. In *S. aureus*, the cation/proton antiporter A (CpaA) is activated by binding c-di-AMP (*Corrigan et al., 2013*). Moreover, c-di-AMP binds to the riboswitch upstream of the genes encoding the K$^+$ transporters and controls their transcriptional levels (*Wang et al., 2019*). With these mechanisms, osmotic stress is controlled by c-di-AMP. Of note, *Chlamydia* lacks annotated orthologs of K$^+$ transporters. In addition to osmotic homeostasis, c-di-AMP has been reported to affect DNA replication and sporulation (*Bejerano-Sagie et al., 2006*). When DNA damage is detected, DisA, a diadenylate cyclase, forms a DNA repair complex with RecA and RadA, resulting in inhibition of diadenylate cyclase activity in *B. subtilis*. Subsequently, c-di-AMP levels decrease, and DNA replication and sporulation are arrested (*Bejerano-Sagie et al., 2006*; *Oppenheimer-Shaanan et al., 2011*; *Zhang and He, 2013*; *Gándara et al., 2017*).

Given the differences in c-di-AMP levels between EBs and RBs and its function as a diffusible second messenger signaling molecule, we hypothesized that the accumulation of c-di-AMP during the developmental cycle might be a trigger for secondary differentiation in *Chlamydia* (*Figure 1B*). Once a threshold concentration of c-di-AMP has been reached in a given RB, it will begin the differentiation process to an EB. Importantly, we do not propose that c-di-AMP is necessarily required for RB replication or growth. We predicted that, if our hypothesis were correct, then we should be able to alter the levels of c-di-AMP in the organism and affect its differentiation kinetics accordingly. For example, if we increase c-di-AMP production, then we anticipate prematurely triggering RB-to-EB conversion with a concomitant reduction in overall growth and replication (since only RBs divide). Conversely, if we prevent c-di-AMP production, then we anticipate delaying EB production without impacting growth rate (i.e., normal replication with reduced RB-to-EB conversion).

There are three principal mechanisms to regulate c-di-AMP levels: through synthesis by diadenylate cyclase, through degradation by a phosphodiesterase (PDE), and through secretion by a transporter (*Corrigan and Gründling, 2013*). Interestingly, *C. trachomatis* only encodes the synthesis mechanism as it possesses the genes for diadenylate cyclase (*dacA*) and its regulator (*ybbR*) within a bicistronic operon, but no annotated PDEs or c-di-AMP transporters (*Stephens et al., 1998*). Both DacA and YbbR are predicted to contain transmembrane domains (3 for DacA and 1 for YbbR) (*Figure 1—figure supplement 1*; *Barker et al., 2013*). To test our hypothesis, we genetically manipulated the levels of DacA and/or YbbR using recently developed strategies in the field. In the present study, we characterized the growth and developmental cycle state of chlamydial strains producing high or low levels of c-di-AMP. Our data show that higher levels of c-di-AMP are directly linked to increased transcript levels of late genes associated with secondary differentiation as well as the concomitant production of EBs at an earlier stage in the developmental cycle. In contrast, in cells with reduced c-di-AMP levels, chlamydial growth was impaired, and the developmental cycle was significantly delayed. Based on these data, we conclude that there is a threshold level of c-di-AMP necessary to trigger secondary differentiation in *C. trachomatis*. This is the first study to identify a physiological function for c-di-AMP in *Chlamydia* as well as a signaling mechanism by which secondary differentiation is initiated in these unique bacteria.

# Results

## Cyclic di-AMP levels increase at later stages of the developmental cycle and in bacteria that overexpress both DacA and YbbR

To verify that EBs have higher levels of c-di-AMP compared to RBs, we infected HeLa cells at a multiplicity of infection (MOI) of 1 with a transformant of *C. trachomatis* L2 carrying a shuttle plasmid with inducible mCherry. This strain serves as a control for subsequent overexpression experiments, and we induced mCherry expression with 5 nM aTc at 10 hr post-infection (hpi). We collected uninfected and infected HeLa cell lysates at 16 and 24 hpi and measured c-di-AMP levels by ELISA (*Figure 1C*). As expected, only basal amounts (<100 pg/mL) of c-di-AMP were detected in uninfected HeLa cells at either timepoint (*Figure 1C*). In infected cells, the 16 hpi timepoint is characterized by predominantly an RB population whereas the 24 hpi timepoint is characterized by ongoing secondary differentiation and a mixture of RBs, IBs, and EBs. We observed that c-di-AMP levels were significantly higher at 24 hpi (~1700 pg/mL) compared to 16 hpi (~250 pg/mL) and increased approximately sevenfold over this timeframe (*Figure 1C*). These results are in agreement with the Barker et al. study indicating higher levels of c-di-AMP in EBs (*Barker et al., 2013*). From these data, we infer that the concentration of c-di-AMP in our culture conditions necessary to trigger conversion of RBs to EBs is at most 1700 pg/ mL.

To test the link between c-di-AMP levels and chlamydial developmental cycle progression, we made a collection of *C. trachomatis* strains carrying anhydrotetracycline (aTc)-inducible constructs (*Figure 1D*). These included strains to overexpress a wild-type or a catalytically dead (D164N) DacA isoform (*Girardin et al., 2003*), wild-type or mutant isoforms of DacA lacking transmembrane domains (ΔTM), YbbR_6xH, or both wild-type or mutant DacA and YbbR_6xH (*dacA*op or *dacA*opMut). For all of these overexpression strains, we emphasize that the chromosomal expression of wild-type *dacA* and *ybbR* is maintained. In addition to the overexpression constructs, we also made a conditional knockdown construct for *dacA* (*dacA*-KD), targeting its promoter region, to decrease the expression of *dacA-ybbR* using a dCas12/crRNA-based CRISPRi system that our group developed for *Chlamydia* (*Ouellette et al., 2021*). Finally, we made a complementation construct for *dacA*-KD by introducing *dacA-ybbR*_6xH (*dacA*-KDcom) 3′ to dCas12 such that induction of dCas12 results in the coexpression of DacA and YbbR_6xH during knockdown of endogenous *dacA-ybbR* transcripts.

To assess whether we could alter c-di-AMP levels in our various strains, we first measured c-di-AMP levels from infected cell lysates at 16 and 24 hpi after inducing overexpression or knockdown of the target genes at 10 hpi (*Figure 1E*). Overexpressing DacA resulted in ~ fourfold increase in c-di-AMP levels as compared to the mCherry-expressing strain at 16 hpi (~1,000 pg/mL) and ~ threefold increase at 24 hpi (~5000 pg/mL). Of note, the level of c-di-AMP in the DacA overexpressing condition at 16 hpi was below the 24 hpi level for EBs (EB threshold line in graphs: ~1700 pg/mL). Overexpressing YbbR_6xH did not impact c-di-AMP production. When overexpressing the catalytically inactive mutant of DacA(D164N), c-di-AMP levels were ~ twofold lower at 16 hpi (~115 pg/mL) and, at 24 hpi (~320 pg/mL), was similar to the 16 hpi mCherry control, indicating a severe reduction in c-di-AMP accumulation. Overexpressing ΔTMDacA phenocopied the effect of overexpressing full-length DacA on c-di-AMP production. Interestingly, the negative effects of overexpressing full-length DacA(D164N) associated with reduced c-di-AMP levels were lost when expressing ΔTMDacA(D164N), which phenocopied the mCherry-expressing control.

When DacA and YbbR_6xH were co-overexpressed, c-di-AMP levels increased by approximately 30-fold at 16 hpi (~8000 pg/mL) and 120-fold at 24 hpi (~340,000 pg/mL) compared to that of the mCherry-expressing control strain. Of note, the levels of c-di-AMP at 16 hpi in the *dacA*op strain (~8000 pg/mL) were higher even than the control strain at 24 hpi (~1700 pg/mL; EB threshold in *Figure 1E*). To confirm whether DacA enzyme activity is critical for the high levels of c-di-AMP in the *dacA*op strain, we co-overexpressed DacA(D164N) and YbbR_6xH (*dacA*opMut). Here, c-di-AMP levels were reduced compared to the control at 16 (~100 pg/mL) and 24 hpi (~400 pg/mL), suggesting that c-di-AMP synthase activity was blocked in this strain similar to overexpressing the DacA(D164N) alone.

For the *dacA*-KD strain, the c-di-AMP level was unchanged at 16 hpi and reduced a statistically significant ~ fivefold at 24 hpi (~500 pg/mL; roughly twice the level of the 16 hr mCherry-expressing control but below the EB threshold). The loss of c-di-AMP production in this strain was restored in the complemented *dacA*-KDcom strain, which showed a phenotype similar to the *dacA*op overexpression

strain with c-di-AMP levels above the EB threshold at both timepoints assessed (~3700 pg/mL at 16 hpi; ~10,000 pg/mL at 24 hpi). Based on these data, we conclude that both DacA and YbbR are necessary for optimal c-di-AMP synthesis in *Chlamydia*. Importantly, our collection of strains that are high or low producers of c-di-AMP give us an opportunity to test our overarching hypothesis (*Figure 1B*).

## Overexpression of membrane-localized DacA isoforms, but not YbbR, disrupts chlamydial growth and development

To begin exploring the impact of altering *dacA* and/or *ybbR* expression on chlamydial growth, we performed a series of experiments to assess their protein localization in chlamydiae. We also measured impacts of overexpressing each individually on chlamydial growth and development. To observe the localization of DacA and YbbR, we infected HeLa cells with transformants encoding *dacA* or *ybbR_6xH* alone and induced expression of the constructs at 10 hpi with 5 nM aTc. At 24 hpi, infected cells were fixed, and we performed an indirect immunofluorescence assay (IFA) by labeling the chlamydial major outer membrane protein (MOMP) and DacA or 6xH. We observed that both DacA and YbbR localized at the bacterial membrane as expected (*Figure 2A* and *Figure 2—figure supplement 1*). When DacA was overexpressed, the inclusion size was significantly reduced in area (~ sixfold) with larger individual organisms (~ twofold) than those in the uninduced control (*Figure 2—figure supplement 2*). Similarly, overexpressing the inactive DacA(D164N) mutant resulted in smaller inclusions similar to overexpressing wild-type DacA (*Figure 2B*). We did not quantify effects of YbbR_6xH overexpression on inclusion or bacterial size since the measured phenotypes indicated no differences from the uninduced control (see below; *Figure 2—figure supplement 3*).

To investigate the effects of DacA isoforms or YbbR_6xH overexpression on chlamydial growth, we measured EB progeny production using an inclusion-forming unit (IFU) assay and genomic DNA copy number (a proxy for total bacteria, i.e., EB+RB) by qPCR. To quantify IFUs, a lysate from a primary infection is prepared and used to infect a fresh monolayer of cells. Any inclusions in the secondary infection are derived from viable EBs present in the primary infection. The vector control strain showed no differences in IFUs when overexpressing mCherry (*Figure 2—figure supplement 4*). At 24 hpi, both IFUs and genome copy numbers were significantly decreased in the DacA overexpression strain (*Figure 2C and D*). We obtained similar results when overexpressing wild-type DacA with a 6xHis tag in wild-type or STING KO cells, indicating that a C-terminal tag does not impact these phenotypes in *Chlamydia* (*Figure 2—figure supplement 5* and *Figure 2—figure supplement 6*). We also performed the same experiments with the inactive isoform of DacA, DacA(D164N). Like overexpression of the wild-type DacA, overexpression of DacA(D164N) also negatively affected IFU production and resulted in reduced genome copy numbers even though c-di-AMP levels were reduced under these conditions (*Figure 2E and F*). YbbR_6xH overexpression did result in a statistically significant, ~ twofold decrease in IFU production (*Figure 2—figure supplement 3*). This reflects one division cycle difference from the uninduced control and, in the absence of any other phenotypic discrepancy, is not considered biologically relevant. No differences were noted in genome copy number when overexpressing YbbR_6xH (*Figure 2—figure supplement 3*).

To further investigate impacts of overexpression of these proteins on developmental cycle progression, we quantified transcripts by RT-qPCR for a canonical early-cycle gene, *euo*, and two late-cycle genes, *hctA* (an 'early' late gene *Chiarelli et al., 2020*) and *omcB* (a canonical late gene *Belland et al., 2003*). Consistent with the genome copy numbers and IFU data, overexpression of DacA resulted in elevated *euo* transcripts and a reduction in the amounts of the late gene transcripts at 24 hpi (*Figure 2G*). Similar effects on these transcripts were noted when overexpressing DacA(D164N) (*Figure 2H*). We measured no effect of YbbR_6xH overexpression on transcript levels for *euo*, *hctA*, or *omcB* (*Figure 2—figure supplement 3*).

Given that overexpression of either the wild-type or mutant isoform of DacA gave the same phenotype yet yielded differences in c-di-AMP levels, we next explored the need for DacA or DacA(D164N) to be membrane-localized to effect these changes. Therefore, we evaluated the effects of overexpression of wild-type or mutant DacA lacking its transmembrane domains (ΔTM). As noted in *Figure 1*, overexpression of wild-type ΔTM DacA yielded the same level of c-di-AMP as overexpression of the full-length wild-type DacA. In contrast, c-di-AMP levels measured from chlamydiae overexpressing the mutant ΔTM DacA(D164N) were the same as the mCherry-expressing strain but reduced in chlamydiae overexpressing the full-length DacA(D164N). By IFA, both ΔTM isoforms localized to the

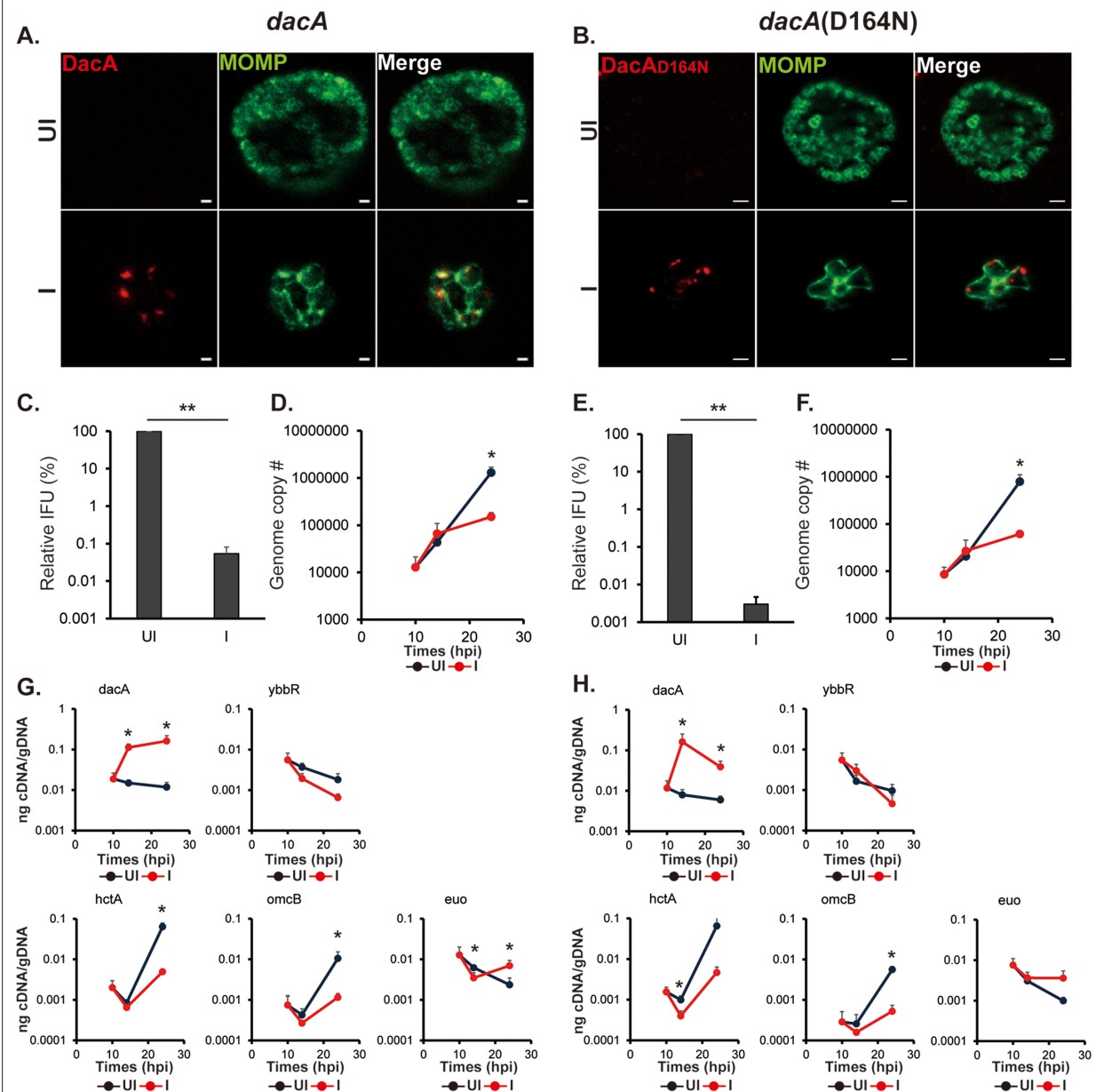

**Figure 2.** Overexpression of DacA or DacA(D164N) is detrimental to the chlamydial developmental cycle. HeLa cells were infected with *C. trachomatis* transformed with a plasmid encoding an anhydrotetracycline (aTc)-inducible DacA or DacA(D164N) (i.e., *dacA* or *dacA*(D164N), respectively; see *Figure 1D*). At 10 hpi, expression of the construct was induced or not with 5 nM aTc, and DNA and RNA samples were collected at 10, 14, and 24 hpi. Immunofluorescence analysis (IFA) and inclusion-forming units (IFU) samples were collected at 24 hpi. For IFA images, the green color represents chlamydial major outer membrane protein (MOMP), which shows the chlamydial cell morphology, and the red color represents DacA or DacA(D164N). (**A** &**B**) IFA images of the *dacA*(**A**) and *dacA*(D164N) (**B**) strains at 24 hpi. Shown are individual panels of a representative inclusion for the strains with DacA and MOMP labeling as well as the merged image. IFA images were acquired on a Zeiss AxioImager.Z2 equipped with an Apotome2 using a 100 X lens objective. Scale bar: 1 µm (**A**) or 2 µm (**B**). (**C** &**D**) Quantification of IFUs (**C**) and genomic DNA copy number (**D**) from uninduced and induced samples of *dacA* at 24 hpi. (**E** &**F**) Quantification of IFUs (**E**) and genomic DNA copy number (**F**) from uninduced and induced samples of *dacA*(D164N) at 24 hpi. (**G** &**H**) Quantification of transcripts by RT-qPCR for *dacA, ybbR, euo, hctA,* and *omcB* from uninduced and induced samples of *dacA* (**G**) and *dacA*(D164N) (**H**) UI = uninduced (i.e. -aTc); I=induced (i.e. +aTc) for all sample types. N=3. *p<0.05; **p<0.001 via two-sample equal variance t-test.

The online version of this article includes the following source data and figure supplement(s) for figure 2:

**Source data 1.** Source data for experiments shown in *Figure 2*.

**Figure supplement 1.** The localization of DacA_6xH (**A**) or YbbR_6xH (**B**) in the first dividing cells.

**Figure supplement 2.** Inclusion area and cell diameter measurements of *dacA* and *dacA*(D164N) expressing strains.

*Figure 2 continued on next page*

*Figure 2 continued*

**Figure supplement 2—source data 1.** Source data for experiments shown in *Figure 2—figure supplement 2*.

**Figure supplement 3.** Overexpression of YbbR_6xH does not affect the chlamydial developmental cycle.

**Figure supplement 3—source data 1.** Source data for experiments shown in *Figure 2—figure supplement 3*.

**Figure supplement 4.** Overexpressed mCherry from the vector control does not affect elementary body (EB) progeny production during the chlamydial developmental cycle.

**Figure supplement 4—source data 1.** Source data for experiments shown in *Figure 2—figure supplement 4*.

**Figure supplement 5.** Overexpression of DacA_6xH is detrimental to the chlamydial developmental cycle.

**Figure supplement 5—source data 1.** Source data for experiments shown in *Figure 2—figure supplement 5*.

**Figure supplement 6.** Quantification of transcripts by RT-qPCR for *dacA, ybbR*, *euo*, *hctA*, and *omcB* from *dacA*-KD and *dacA*_6xH strains cultured in STING-KO HeLa cells.

**Figure supplement 6—source data 1.** Source data for experiments shown in *Figure 2—figure supplement 6*.

cytosol, and there were no observable differences in inclusion or bacterial morphology as compared to the uninduced control (*Figure 3A and B*). We did not quantify effects of overexpression on inclusion or bacterial size since the measured phenotypes indicated no differences from the uninduced control. Overexpression of the ΔTM isoforms resulted in no statistical differences in IFU production or genome copy numbers, and, similarly, there were no differences in transcript levels for *ybbR, euo, hctA,* or *omcB* as compared to the uninduced controls (*Figure 3C–H*). These data indicate that the negative impacts of DacA overexpression are linked to its membrane localization and are independent of c-di-AMP production.

## A low level of c-di-AMP decreases the transcript levels of late genes

As mentioned above, the *dacA*-KD strain displayed a lower c-di-AMP level compared to that of the control strain at 24 hpi, whereas the *dacA*-KDcom complemented strain exhibited high levels of c-di-AMP at this timepoint and at 16 hpi (*Figure 1E*). The complemented strain encodes not only the *dacA*-knockdown system but also the *dacA-ybbR_6xH* operon as a transcriptional fusion with dCas12. To observe effects of knockdown or complementation on the developmental cycle, we first examined inclusion and bacterial morphology by IFA. When the knockdown system was induced, bacterial cell size was enlarged (~ twofold), but we observed no change in inclusion area as compared to that of the uninduced control (*Figure 4A*, *Figure 4—figure supplement 1*). In the complemented strain, induction of dCas12 and YbbR_6xH was confirmed by IFA under inducing conditions (*Figure 4B*). We expected that co-expressing DacA and YbbR_6xH would complement the *dacA*-KD to the wild-type phenotype. However, the inclusion area was reduced (~ threefold) with slightly larger organisms (~1.4 fold) as compared to the uninduced control (*Figure 4—figure supplement 1*), indicating this is not the case.

In addition, we also measured the amount of EB progeny (IFUs) and genome copy numbers. Although genome copy numbers were the same, IFUs decreased by approximately 80% compared to that of the uninduced sample at 24 hpi in the knockdown strain (*Figure 4C and D*), suggesting more RBs are present in the sample. IFUs and genome copies showed an approximate twofold reduction at 24 hpi in the complemented strain (*Figure 4E and F*). The data for the knockdown strain suggest that low levels of c-di-AMP are detrimental for secondary differentiation. As genome copies alone cannot assess the ratio of RBs to EBs, we performed RT-qPCR to quantify transcripts of relevant gene targets. When dCas12 was induced, both *dacA* and *ybbR* transcript levels decreased (*Figure 4G*). Since *dacA* and *ybbR* are transcribed in an operon, this result is not surprising. Consistent with reduced EB yields, transcripts of the late genes *omcB* and *hctA* were decreased at 24 hpi. In contrast, transcripts of the early gene *euo* were slightly elevated at 24 hpi. We have previously observed no effects on genome levels or transcription of these genes when overexpressing the dCas12 gene alone (*Ouellette et al., 2021*), indicating these effects are specific to *dacA*-KD and that *dacA* knockdown reduces or delays expression of late gene transcripts. For the complemented strain, both *dacA* and *ybbR* transcripts were increased compared to the *dacA*-KD strain (*Figure 4H*), and transcript levels for these genes were increased beyond the 'wild-type' uninduced control levels, again suggesting the 'complemented' strain did not return to a wild-type phenotype. Nonetheless, transcripts for the early gene *euo* were indistinguishable to levels measured in the uninduced strain (*Figure 4H*). Surprisingly, *hctA* transcripts

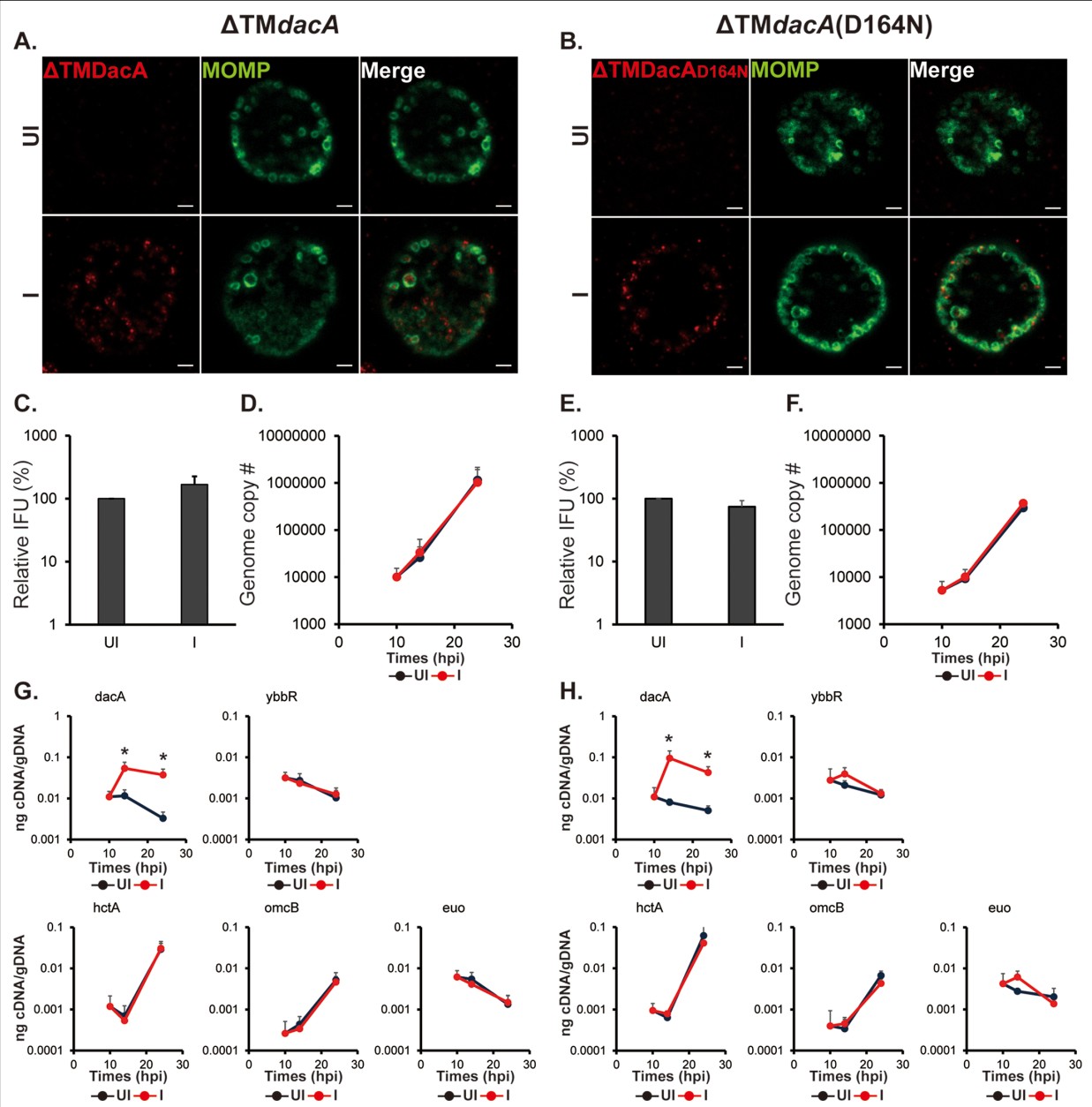

**Figure 3.** Overexpression of ΔTMDacA or ΔTMDacA(D164N) does not alter the chlamydial developmental cycle. HeLa cells were infected with *C. trachomatis* transformed with a plasmid encoding an anhydrotetracycline (aTc)-inducible ΔTMDacA or ΔTMDacA(D164N) (i.e. ΔTM*dacA* and ΔTM*dacA* (D164N); see *Figure 1D*). At 10 hpi, expression of the construct was induced or not with 5 nM aTc, and DNA and RNA samples were collected at 10, 14, and 24 hpi. Immunofluorescence analysis (IFA) and inclusion-forming units (IFU) samples were collected at 24 hpi. For IFA images, the green color represents chlamydial major outer membrane protein (MOMP), which shows the chlamydial cell morphology, and the red color represents ΔTMDacA or ΔTMDacA(D164N). (A & B) IFA images of ΔTM*dacA*(A) and ΔTM*dacA*(D164N)(B) strains at 24 hpi. Shown are individual panels of a representative inclusion for the strains with DacA and MOMP labeling as well as the merged image. IFA images were acquired on a Zeiss AxioImager.Z2 equipped with an Apotome2 using a 100 X lens objective. Scale bar: 2 μm. (C & D) Quantification of IFUs (C) and genomic DNA copy number (D) from uninduced and induced samples of ΔTM*dacA* at 24 hpi. (E & F) Quantification of IFUs (E) and genomic DNA copy number (F) from uninduced and induced samples of ΔTM*dacA*(D164N). (G & H) Quantification of transcripts by RT-qPCR for *dacA, ybbR*, *euo, hctA*, and *omcB* from uninduced and induced samples of ΔTM*dacA* (G) and ΔTM*dacA*(D164N) (H) UI = uninduced (i.e. -aTc); I=induced (i.e. +aTc) for all sample types. N=3. *p<0.05 via two-sample equal variance t-test.

The online version of this article includes the following source data for figure 3:

**Source data 1.** Source data for experiments shown in *Figure 3*.

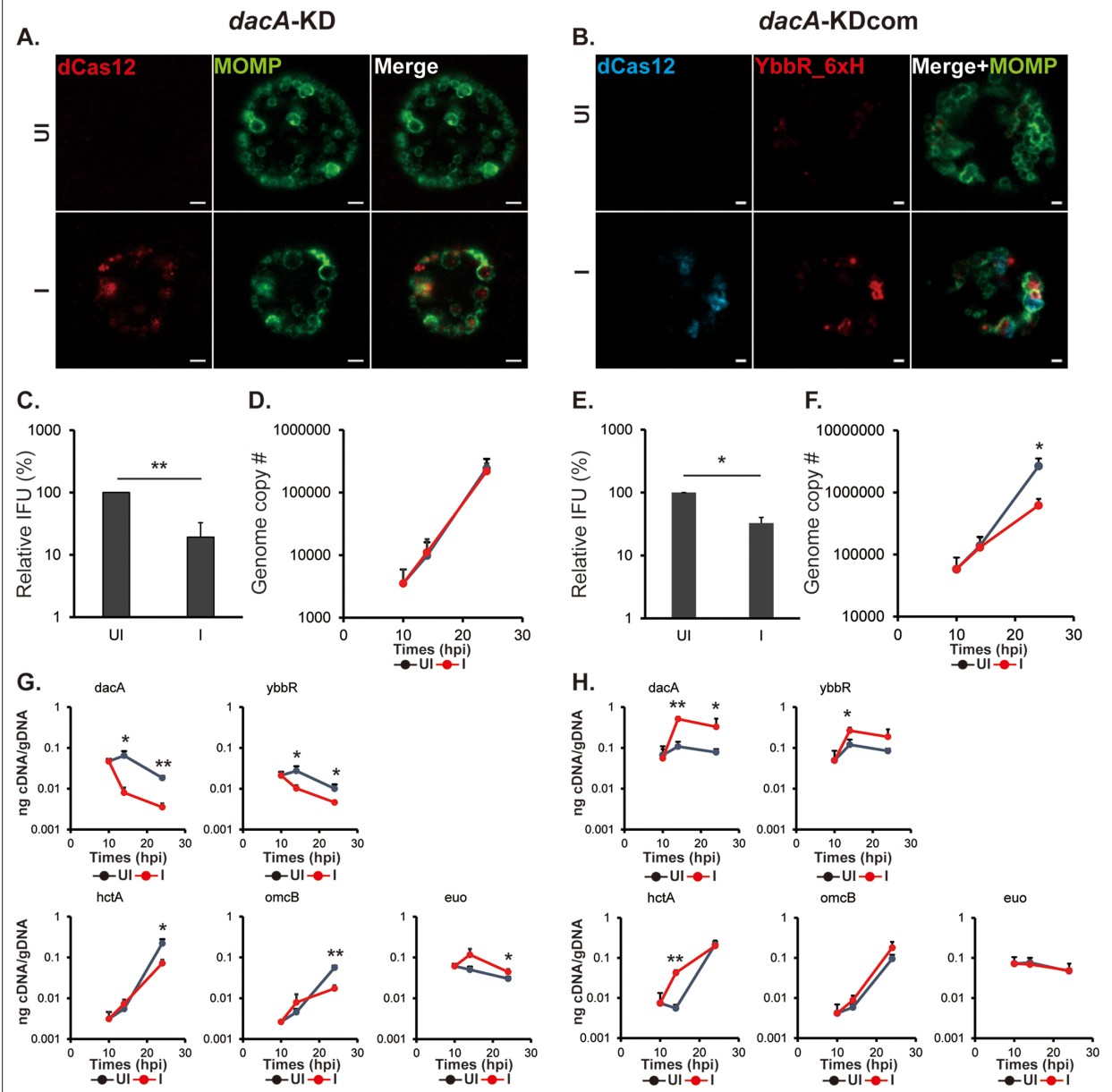

**Figure 4.** CRISPRi-mediated *dacA-ybbR* knockdown displays reduced levels of transcripts for late genes. HeLa cells were infected with *C. trachomatis* transformed with a plasmid encoding an anhydrotetracycline (aTc)-inducible CRISPRi-dCas12 system targeting the *dacA* promoter (*dacA*-KD) or *dacA*-KD system and DacA/YbbR_6xH (i.e., *dacA*-KDcom; see *Figure 1D*). At 10 hpi, knockdown was induced or not with 5 nM aTc, and DNA and RNA samples were collected at 10, 14, and 24 hpi. Immunofluorescence analysis (IFA) and inclusion-forming units (IFU) samples were collected at 24 hpi. (**A** & **B**) IFA images of the *dacA*-KD (**A**) and *dacA*-KDcom (**B**) strains at 24 hpi. Shown are individual panels of a representative inclusion for the strains for dCas12, YbbR_6xH, and major outer membrane protein (MOMP) labeling as well as the merged image. IFA images were acquired on a Zeiss AxioImager.Z2 equipped with an Apotome2 using a 100 X lens objective. Scale bar: 2(**A**) or 1(**B**) μm. (**C** & **D**) Quantification of IFUs (**C**) and genomic DNA copy number (**D**) from uninduced and induced samples of *dacA*-KD at 24 hpi. (**E** and **F**) Quantification of IFUs (**E**) and genomic DNA copy number (**F**) in uninduced and induced samples of *dacA*-KDcom. (**G** & **H**) Quantification of transcripts by RT-qPCR for *dacA, ybbR, euo, hctA,* and *omcB* from uninduced and induced samples of *dacA*-KD (**G**) and *dacA*-KDcom (**H**) UI = uninduced (i.e. -aTc); I=induced (i.e. +aTc) for all sample types. N=3. *p<0.05; **p<0.001 via two-sample equal variance t-test.

The online version of this article includes the following source data and figure supplement(s) for figure 4:

**Source data 1.** Source data for experiments shown in *Figure 4*.

**Figure supplement 1.** Inclusion area and cell diameter measurements of *dacA*-KD and *dacA*-KDcom.

**Figure supplement 1—source data 1.** Source data for experiments shown in *Figure 4—figure supplement 1*.

were increased ~10 fold at 14 hpi compared to that of the uninduced sample, and *omcB* transcripts were slightly, but not significantly, increased at 14 and 24 hpi (*Figure 4H*). These data suggest that overexpressing DacA and YbbR_6xH, with associated increases in c-di-AMP levels (*Figure 1*), may alter the timing of secondary differentiation.

## High levels of c-di-AMP induce late gene expression

To clarify the effect of DacA and YbbR_6xH overexpression on secondary differentiation, we next evaluated the phenotype of the *dacA*op strain in contrast with the *dacA*opMut strain, in which the active site residue of DacA has been mutated. When we induced expression of the wild-type constructs, we confirmed the induction of DacA and YbbR_6xH and their colocalization at the membrane *Figure 5A*; Pearson correlation coefficient of 0.713±0.109 from 20 inclusions measured by JACoP Plugin of ImageJ; values near 1 indicate colocalization (*Bolte and Cordelières, 2006*; *Dunn et al., 2011*). This is not surprising as both proteins are critical for c-di-AMP synthesis based on our c-di-AMP measurements (*Figure 1E*). Organism and inclusion morphology were similar to the *dacA*-KDcom complemented strain (*Figures 4B and 5A*; *Figure 5—figure supplement 1*). The bacterial morphology of the *dacA*opMut strain indicated larger organisms (~1.8 fold) in smaller inclusions (~2.5 fold) after inducing expression (*Figure 5B*; *Figure 5—figure supplement 1*). We next assessed whether the overexpressed DacA and YbbR_6xH affected IFU production and replication. IFUs were reduced roughly twofold for the *dacA*op strain and 1000-fold in the *dacA*opMut strain after inducing overexpression (*Figure 5C&E*). Again, the genome copy data for *dacA*op overexpression closely phenocopied the complemented knockdown strain, showing a decrease at 24 hpi (*Figure 5D*). Similarly, genome copy numbers for the *dacA*opMut strain showed a significant drop at 24 hpi after inducing expression (*Figure 5F*).

We then quantified transcripts for the developmentally regulated genes *euo*, *hctA*, and *omcB* as well as for *dacA* and *ybbR* (*Figure 5G&H*). Not surprisingly, *dacA* and *ybbR* transcripts were elevated at the timepoints assessed under inducing conditions for both the *dacA*op and *dacA*opMut strains. Transcripts for *euo* were not statistically changed but trended higher at the 24 hpi timepoint, whereas *omcB* transcripts were slightly, but not significantly, increased at 14 and 24 hpi during *dacA*op overexpression. Once again, we observed that *hctA* transcripts were increased over 10-fold at 14 hpi in the developmental cycle (*Figure 5G*), similar to what we measured for the complemented knockdown strain (*Figure 4H*). These data reinforce that elevated c-di-AMP levels (*Figure 1E*) in these strains lead to increased expression of the late gene *hctA* at an earlier timepoint (14 hpi) in the developmental cycle. As further validation of this *dacA*op overexpression strain that was constructed in a beta-lactamase producing background, we also generated a spectinomycin-resistant *dacA*op overexpression strain and validated that induction of the *dacA* operon resulted in earlier accumulation of *hctA* transcripts (*Figure 5—figure supplement 2*). In assessing developmentally regulated transcripts for the *dacA*opMut strain, *euo* levels were maintained, albeit not significantly so, at 24 hpi (*Figure 5H*). In contrast to the c-di-AMP overproducing strains, transcripts for *hctA* and *omcB* were decreased at 24 hpi (*Figure 5H*). The transcriptional results of the developmentally regulated genes in the *dacA*opMut strain were very similar to the *dacA*-KD strain (*Figure 4G*), suggesting that blocking c-di-AMP accumulation interferes with developmental cycle progression.

## Elevated c-di-AMP levels result in increased transcript levels of genes necessary for secondary differentiation

Given the surprising finding that, in strains overproducing c-di-AMP, *hctA* transcripts were 10-fold higher at a timepoint not associated with secondary differentiation, we asked the question whether *all* genes related to secondary differentiation were increased after inducing production of c-di-AMP. Conversely, we wanted to explore whether reducing c-di-AMP levels would delay expression of genes related to secondary differentiation. Thus, to further investigate how c-di-AMP affects transcription of such genes, we performed RNA sequencing on both the *dacA*op overexpression and *dacA*-KD strains and compared the transcriptome between uninduced and induced samples within the given strain at the given timepoint. HeLa cells were infected with these transformants, and overexpression or knockdown was induced or not at 10 hpi with 5 nM aTc. For the *dacA*op strain, RNA was collected at 16 hpi, a time at which late genes are beginning to be expressed (as opposed to 14 hpi) but remain near a basal level of transcription (*Belland et al., 2003*). The rationale for this was to determine whether

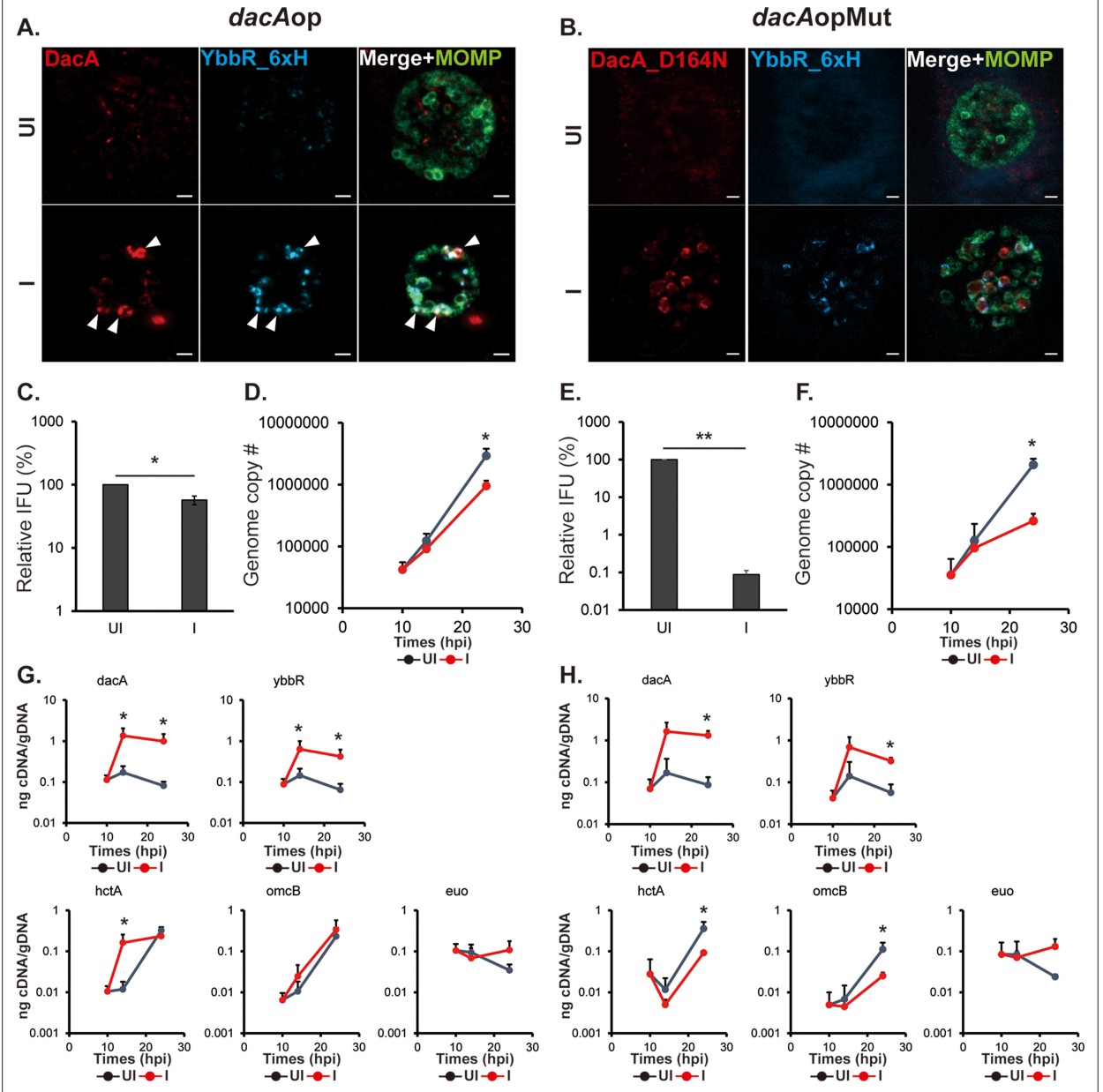

**Figure 5.** Overexpression of DacA and YbbR_6xH prematurely increases *hctA* transcript levels. HeLa cells were infected with *C. trachomatis* transformed with an aTc-inducible plasmid encoding wild-type DacA/YbbR_6xH or DacA(D164N)/YbbR_6xH (i.e., *dacA*op and *dacA*opMut, respectively; see *Figure 1D*). At 10 hpi, expression of the constructs was induced or not with 5 nM aTc, and DNA and RNA samples were collected at 10, 14, and 24 hpi. Immunofluorescence analysis (IFA) samples were collected at 24 hpi. (**A** &**B**). IFA images of the *dacA*op (**A**) and *dacA*opMut (**B**) at 24 hpi. Shown are individual panels of a representative inclusion for the strains for DacA and YbbR_6xH as well as the merged image with major outer membrane protein (MOMP) labeling. The arrowheads represent the co-localization of DacA and YbbR. IFA images were acquired on a Zeiss AxioImager.Z2 equipped with an Apotome2 using a 100 X lens objective. Scale bar: 2 μm. (**C** & **D**) Quantification of IFUs (**C**) and genomic DNA copy number (**D**) from uninduced and induced samples of *dacA*op at 24 hpi. (**E** & **F**). Quantification of IFUs (**E**) and genomic DNA copy number (**F**) from uninduced and induced samples of *dacA*opMut. (**G** & **H**). Quantification of transcripts by RT-qPCR for *dacA, ybbR, euo, hctA,* and *omcB* from uninduced and induced samples of *dacA*op (**G**) and *dacA*opMut (**H**). UI = uninduced (i.e. -aTc); I=induced (i.e. +aTc) for all sample types. N=3. *p<0.05 two-sample equal variance t-test.

The online version of this article includes the following source data and figure supplement(s) for figure 5:

**Source data 1.** Source data for experiments shown in *Figure 5*.

**Figure supplement 1.** Inclusion area and cell diameter measurements of *dacA*op and *dacA*opMut.

**Figure supplement 1—source data 1.** Source data for experiments shown in *Figure 5—figure supplement 1*.

**Figure supplement 2.** Phenotypic characterization of a *dacA*op overexpression construct encoding spectinomycin resistance (Spc$^R$).

*Figure 5 continued on next page*

*Figure 5 continued*

**Figure supplement 2—source data 1.** Source data for experiments shown in *Figure 5—figure supplement 2*.

high c-di-AMP levels result in increased late gene transcripts at this timepoint *above and beyond* the levels of the control, uninduced condition. For the *dacA*-KD strain, RNA was collected at 24 hpi, a time at which late genes are peaking in their expression. The rationale for this was to determine whether late gene transcription was decreased, which could not otherwise be reliably assessed at the 16 hpi timepoint.

RNA sequencing results were statistically analyzed by the UNMC Bioinformatics Core (*Supplementary file 1* and *Supplementary file 2*) . *Figure 6* shows a volcano plot of the results for the *dacA*op overexpression and *dacA*-KD strains. Of note, many late genes were evident in the upregulated quadrant for the *dacA*op strain whereas these genes were present in the downregulated quadrant for the *dacA*-KD strain. We further characterized the upregulated or downregulated gene sets for the *dacA*op overexpression and *dacA*-KD strains, respectively, based on significant difference (p<0.05) and fold-change (>1.5). A summary of these results is presented in *Table 1* (all data are presented in *Supplementary file 2*). We grouped the differentially expressed genes into five categories: (1) canonical late genes for which the literature has associated them with EB function, (2) outer membrane-associated, (3) gene regulation-associated, (4) glycogen synthesis-associated, and (5) type III secretion system associated. Recent work from our group and the Hefty group to define the regulons of the alternative sigma factors in *Chlamydia* demonstrated that these sigma factors regulate

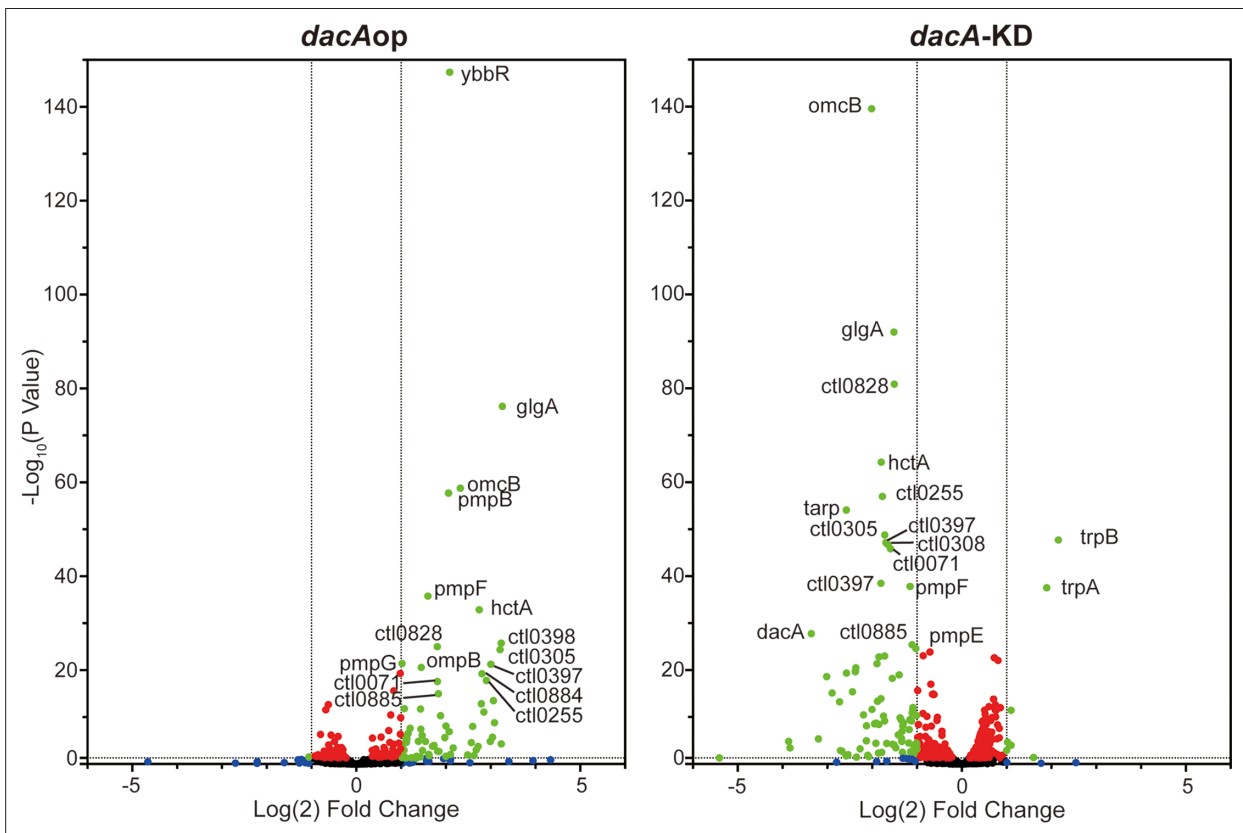

**Figure 6.** The levels of c-di-AMP are correlated with late gene transcripts. RNA sequencing was performed from HeLa cells infected with either the *dacA*op or *dacA*-KD strains. RNA samples were collected at 16 hpi for *dacA*op and 24 hpi for *dacA*-KD after inducing expression of the relevant constructs at 10 hpi. Shown is a volcano plot of the RNA sequencing results with the vertical dashed lines indicating a twofold change in transcript levels as compared to the respective uninduced control for the given strain and the horizontal lines indicating a p-value of 0.05. The plot was made using GraphPad Prism software. Green spots represent genes demonstrating a statistically significant twofold change in transcription levels between uninduced and induced samples. Red dots represent genes with significant changes in transcript levels less than twofold. Blue dots represent genes not significantly different but more than twofold changed between the conditions. Black dots represent genes not significantly different and less than twofold changed between the conditions. See also *Table 1* and *Supplementary file 1* and *Supplementary file 2* for more details.

**Table 1.** Genes impacted by cyclic di-AMP levels.

| | | | Canonical late genes | Fold change | | |
|---|---|---|---|---|---|---|
| Gene ID | Ctr D ORF | Name | Protein names | dacAop_OE | dacA-KD | Reference |
| CTL0112 | CT743 | hctA | Histone H1-like protein HC1 | 6.70 | –3.47 | *Belland et al., 2003*; *Fahr et al., 1995* |
| CTL0302 | CT046 | hct2 | Histone H1-like protein HC2 | 3.52 | –5.96 | *Belland et al., 2003*; *Yu et al., 2006* |
| CTL0336 | CT080 | ltuB | Late transcription unit B protein | 4.00 | –3.61 | *Belland et al., 2003*; *Fahr et al., 1995* |
| CTL0700 | CT441 | tsp | Carboxy-terminal processing protease | 6.18 | –6.62 | *Yu et al., 2006* |
| CTL0702 | CT443 | omcB | Large cysteine-rich periplasmic protein | 5.00 | –4.04 | *Belland et al., 2003*; *Soules et al., 2020b* |
| CTL0703 | CT444 | omcA | Small cysteine-rich outer membrane protein | 5.93 | –3.71 | *Belland et al., 2003*; *Soules et al., 2020b* |
| CTL0716 | CT456 | tarp | Translocated actin-recruiting phosphoprotein | 6.06 | –5.97 | *Hatch and Ouellette, 2023*; *Soules et al., 2020b* |
| | | | **Membrane organization associated** | **Fold change** | | |
| Gene ID | Ctr D ORF | Name | Protein Names | dacAop_OE | dacA-KD | Reference |
| CTL0082 | CT713 | ompB | Outer membrane protein B | 2.73 | –2.15 | *Nicholson et al., 2003* |
| CTL0248 | CT869 | pmpE | Polymorphic outer membrane protein | 1.99 | –2.04 | *Hatch and Ouellette, 2023*; *Nicholson et al., 2003* |
| CTL0249 | CT870 | pmpF | Polymorphic outer membrane protein | 3.03 | –2.23 | *Hatch and Ouellette, 2023*; *Nicholson et al., 2003* |
| CTL0250 | CT871 | pmpG | Polymorphic outer membrane protein | 2.03 | –1.55 | *Belland et al., 2003*; *Nicholson et al., 2003* |
| CTL0429 | CT177 | dsbA | Disulfide bond chaperone | 2.79 | –2.47 | *Hatch and Ouellette, 2023* |
| CTL0610 | CT356 | dsbH | Thioredox_DsbH domain-containing protein | 3.07 | –2.11 | *Belland et al., 2003*; *Nicholson et al., 2003* |
| CTL0670 | CT413 | pmpB | Polymorphic outer membrane protein | 4.17 | –1.74 | *Hatch and Ouellette, 2023*; *Nicholson et al., 2003* |
| | | | **Gene Regulation associated** | **Fold change** | | |
| Gene ID | Ctr D ORF | Name | Protein Names | dacAop_OE | dacA-KD | Reference |
| CTL0044 | CT675 | mcsB | Protein-arginine kinase | 3.52 | –2.21 | *Hatch and Ouellette, 2023* |
| CTL0045 | CT676 | mcsA | UVR domain-containing protein | 2.53 | –2.92 | *Hatch and Ouellette, 2023* |
| CTL0727 | CT467 | atoS | Two component regulator, histidine kinase | 3.33 | –1.80 | *Hatch and Ouellette, 2023* |
| CTL0728 | CT468 | atoC | Two-component system response regulator | 7.95 | –2.78 | |
| CTL0894 | CT630 | chxR | Atypical response regulator protein ChxR | 3.68 | –2.93 | *Yang et al., 2017* |
| | | | **Glycogen synthesis** | **Fold change** | | |
| Gene ID | Ctr D ORF | Name | Protein Names | dacAop_OE | dacA-KD | Reference |
| CTL0167 | CT798 | glgA | Glycogen synthase | 9.58 | –2.86 | *Belland et al., 2003*; *Nicholson et al., 2003* |
| CTL0342 | CT087 | malQ | 4-alpha-glucanotransferase | 9.43 | –4.29 | *Hefty and Stephens, 2007* |
| CTL0500 | CT248 | glgP | Alpha-1,4 glucan phosphorylase | 2.70 | –1.71 | *Nicholson et al., 2003* |
| | | | **Type III Secretion System** | **Fold change** | | |
| Gene ID | Ctr D ORF | Name | Protein Names | dacAop_OE | dacA-KD | Reference |
| CTL0041 | CT672 | sctQ | Type III secretion component, basal body | 1.93 | –1.76 | *Hefty and Stephens, 2007* |
| CTL0043 | CT674 | cdsC | Type III secretion structural protein | 1.79 | –1.56 | *Hefty and Stephens, 2007* |
| CTL0343 | CT088 | scc1 | Type III secretion chaperone | 4.48 | –3.78 | *Hefty and Stephens, 2007* |
| CTL0824 | CT561 | sctL | Type III secretion system protein | 2.18 | –1.75 | *Hatch and Ouellette, 2023* |

OE = overexpression.
KD = knockdown.

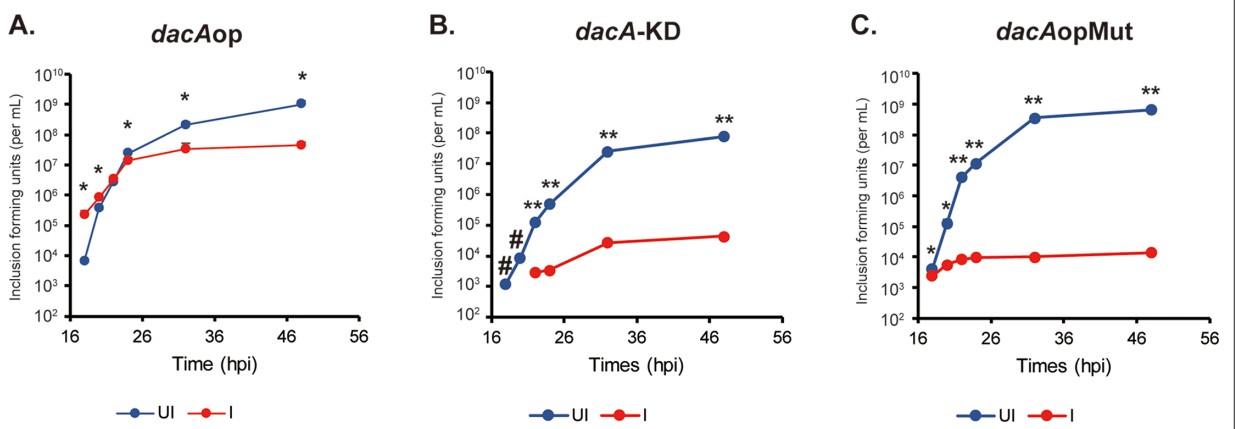

**Figure 7.** Elementary body (EB) production is induced by high levels of c-di-AMP. (**A–C**) HeLa cells were infected with the chlamydial transformants *dacA*op (**A**), *dacA*-KD (**B**), or *dacA*opMut (**C**). Expression of the constructs was induced or not with 5 nM aTc at 10 hpi. At 18, 20, 22, 24, 32, and 48 hpi, infected cell lysates were harvested for inclusion-forming unit (IFU) quantification. UI = uninduced (i.e. -aTc); I=induced (i.e. +aTc) for all sample types. #: Non-detected from the induced samples. N=3. *p<0.05; **p<0.001 via two-sample equal variance t-test.

The online version of this article includes the following source data for figure 7:

**Source data 1.** Source data for experiments shown in *Figure 7*.

some late gene expression associated with outer membrane remodeling, type III secretion, and other processes (*Hatch and Ouellette, 2023*; *Soules et al., 2020b*). Consistent with our RT-qPCR data, we observed that all canonical late genes, as well as all the other genes listed in these categories, showed an increase in expression after c-di-AMP production was induced. In contrast, all the genes, and particularly the canonical late genes, showed a decrease in expression under conditions where c-di-AMP production was impaired. Overall, these RNA sequencing data confirm the direct influence of c-di-AMP on expression of genes related to secondary differentiation.

## Alterations in c-di-AMP levels impact the timing of EB production

The phenotypic effect of increased late gene transcription should be an increase in EB production. However, if EB production is initiated at an earlier time in the developmental cycle when there are fewer non-infectious RBs to convert, then overall EB production should be decreased with a concomitant decrease in genome copies since only the RB replicates DNA. Conversely, delayed expression of late genes should be associated with a delay in EB production. Therefore, to test these predictions, we quantified EB production at 2 hr intervals from 18 to 24 hpi to assess EB production during earlier phases of the developmental cycle, as well as at 32 and 48 hpi to assess overall EB yields (*Figure 7*). We did not detect any EBs at 16 hpi or earlier (not shown). Cells were infected with the *dacA*op, *dacA*opMut, and *dacA*-KD strains and induced or not at 10 hpi with 5 nM aTc as previously noted. Consistent with the transcriptional data, we measured higher EB yields at 18 and 20 hpi during *dacA*op overexpression that quickly plateaued by 24 hpi (*Figure 7A*). Conversely, when c-di-AMP accumulation was blocked by reducing the expression of DacA and YbbR (*dacA*-KD), we observed delayed EB production (*Figure 7B*). We also measured lower EB yields when c-di-AMP levels were decreased by overexpression of the DacA(D164N) isoform and YbbR (*dacA*opMut; *Figure 7C*). The uninduced control conditions for each strain showed a steady accumulation of EBs throughout the course of the experiment, as expected. The vector control strain expressing mCherry showed no differences as previously noted (*Figure 2—figure supplement 4*). From these data, we conclude that earlier production of c-di-AMP results in earlier production of EBs.

## Discussion

Secondary differentiation is essential for the propagation and survival of *Chlamydia* species. However, the mechanisms governing this essential process are largely unknown. It is possible that both internal and external environmental changes might serve as signals to trigger the shift from the non-infectious

RB to the infectious EB. It is also possible, and even likely, that *Chlamydia* integrates multiple signal inputs during the differentiation process. For example, we recently described two post-translational processes that impact secondary differentiation: the activity of the unfoldases ClpX and ClpC (*Jensen et al., 2025*; *Wood et al., 2022*) and the redox status of *Chlamydia* (*Singh and Ouellette, 2025*). As secondary differentiation is asynchronous, any mechanism must account for the stochasticity within the population of RBs 'considering' differentiating to EBs. Here, we provide the first evidence of a signaling molecule that can directly activate EB-associated gene expression with a concomitant early production of EBs.

Several reports have proposed mechanisms by which *Chlamydia* triggers secondary differentiation. For example, Thompson et al. investigated the non-canonical regulation of the major sigma factor, $\sigma^{66}$, by the Rsb phosphoregulatory system (*Thompson et al., 2015*). The authors provided evidence that altering the levels of RsbW or RsbV1 impacted the expression of $\sigma^{66}$-controlled genes. This led them to propose a model whereby sensing of ATP by the Rsb system controls the availability of $\sigma^{66}$, implying that low levels of ATP lead to initiation of secondary differentiation. Further support from this came from work by Kuwabara et al. who showed effects of ATP, GTP, and glucose on the activity of different Rsb components (*Kuwabara et al., 2022*). Finally, work from Soules et al. identified TCA intermediates as ligands for RsbU, linking the TCA cycle and ATP synthesis to the Rsb system (*Soules et al., 2020a*). However, no effect was noted on premature late gene expression in these experimental systems.

Sequestration/inactivation of the major sigma factor under low ATP conditions would presumably render RNA polymerase free to interact with the alternative sigma factors, $\sigma^{54}$ and $\sigma^{28}$. These alternative sigma factors have been linked to late gene expression with studies from Soules et al. and Hatch and Ouellette indicating that $\sigma^{54}$ is associated with increased transcription of outer membrane components, type III secretion (T3S) system components, and other genes typically expressed late in development (*Hatch and Ouellette, 2023*; *Soules et al., 2020b*). However, some of these changes are likely indirect and downstream to the $\sigma^{54}$ regulon itself since some affected genes have been shown to have $\sigma^{66}$ promoter sequence elements (*Ouellette et al., 2005*; *Hefty and Stephens, 2007*). Significant remodeling of the outer membrane occurs as the RB transitions to the EB (*Tamura et al., 1971*; *Tamura and Manire, 1967*), and, similarly, the EB is prepackaged with T3S effectors, like TarP (*Clifton et al., 2004*), that facilitate EB invasion into a target host cell. $\sigma^{28}$ was shown to control expression of two canonical late genes: *hctB* and *tsp* (*Hatch and Ouellette, 2023*). Each of the encoded proteins is highly toxic if overexpressed (*Swoboda et al., 2023*; *Grieshaber et al., 2022*), suggesting that the additional layer of regulation by $\sigma^{28}$ is necessary to ensure EBs are formed at the correct time. Hatch and Ouellette proposed that these genes may be the last to be activated precisely because of this (*Hatch and Ouellette, 2023*). Overall, these data clearly link transcriptional regulation, via the sigma factors, to developmental cycle progression and secondary differentiation.

Despite the clear changes in transcription that occur during the chlamydial developmental cycle, activation of late gene expression alone does not guarantee secondary differentiation. We recently explored the function of the Clp protease systems in chlamydial growth and development (*Wood et al., 2022*). Interestingly, we observed that the inability to degrade SsrA-tagged products prevented functional secondary differentiation with a severe defect in EB production. This was demonstrated in strains unable to degrade SsrA products either by preventing their recognition by ClpX or by altering the SsrA tag to a version that is not degraded efficiently. Nonetheless, late gene transcription was activated in these strains. Therefore, post-translational regulation of secondary differentiation is also critical to the process.

Assuming a post-translational gene regulation model for secondary differentiation, we were interested in exploring known differences between EBs and RBs as a clue to what factors could be serving as a signal for this process. EBs and RBs differ in a number of characteristics, from their function to their morphology to their gene expression. In 2013, a study from Barker et al. added another difference – their relative levels of c-di-AMP with RBs having lower levels than EBs. Although the study from Barker et al. was focused on understanding how IFNβ is activated in *Chlamydia*-infected cells, we were intrigued by this relative difference in c-di-AMP levels. We hypothesized that c-di-AMP production might be a signal for secondary differentiation. We compared chlamydial growth in normal HeLa or STING-KO HeLa cells but could not detect any differences in growth in these cell types (*Figure 4G* & *Figure 2—figure supplement 5* vs *Figure 2—figure supplement 6*), indicating that c-di-AMP

production by *Chlamydia* is unlikely used as means of activating the host cell. Rather, STING activation is an 'accidental' outcome during chlamydial developmental cycle progression. Using recently developed genetic tools for *Chlamydia*, we demonstrated that we could manipulate the levels of c-di-AMP in the bacterium to either block or stimulate its production. Excitingly, we observed that elevated c-di-AMP was linked to earlier secondary differentiation whereas blocking c-di-AMP production prevented EB production (*Figure 7*). Importantly, c-di-AMP production resulted in an increase in late gene transcripts, as noted above for the function of the alternative sigma factors (*Figure 6*). Therefore, c-di-AMP may act as a post-translational mechanism to trigger secondary differentiation. We can also directly conclude from our experiments that the threshold level of c-di-AMP necessary to trigger secondary differentiation in our culture conditions is greater than 1000 pg/mL and less than 2000 pg/mL. We base this on the fact that the 16 hpi level of c-di-AMP in the DacA-overexpressing strain (1000 pg/mL) did not trigger secondary differentiation whereas, during a normal infection (e.g., mCherry-expressing strain) when EBs are being produced at 24 hpi, the c-di-AMP level is ~2000 pg/mL. To our knowledge, this is the first study to identify and provide experimental evidence for a signaling factor involved in differentiation of an obligate intracellular bacterium.

How might c-di-AMP function in *Chlamydia*? This is not readily apparent because *Chlamydia* lacks orthologs of proteins that have been characterized in other bacterial systems to bind c-di-AMP. For example, *Chlamydia* lacks annotated $K^+$ transporters, and it is unknown how these bacteria maintain their osmolarity. However, a recent study suggested that changes in $K^+$ levels affect the chlamydial developmental cycle (*Andrew et al., 2021*). Therefore, c-di-AMP may function as a $K^+$ homeostasis coordinator in *Chlamydia* through as-yet unknown pathways. We did observe altered chlamydial morphology in some of our strains in which we altered c-di-AMP levels, and this may indicate changes in osmostability or effects on cell division. In regard to the latter possibility, in *S. aureus*, DacA activity is inhibited by interacting with GlmM (phosphoglucosamine mutase) (*Tosi et al., 2019*). GlmM produces UDP-GlcNAc that is a precursor in both peptidoglycan and LPS synthesis. Given that *Chlamydia* uses peptidoglycan for cell division, we tested whether DacA and YbbR were localized to the division septum. However, their localization was not associated with the division septum (*Figure 2—figure supplement 1*). In addition, we did not detect an interaction between DacA and GlmM using the Bacterial Adenylate Cyclase-based Two-Hybrid (BACTH) system (data not shown), and the chlamydial GlmM lacks the residues required for DacA interaction. Therefore, we conclude that DacA and YbbR are unlikely to directly impact cell division. However, based on the cell morphological changes in DacA overexpression, we cannot exclude a function for c-di-AMP on cell wall metabolism. Future studies will focus on identifying c-di-AMP binding proteins and characterizing their function in *Chlamydia*. We also cannot exclude a function for c-di-AMP in directly manipulating gene expression by riboswitches (*Nelson et al., 2013*), and further work is necessary to understand how c-di-AMP directly controls chlamydial development.

The use of c-di-AMP as a signal for secondary differentiation presents a 'chicken-or-egg' quandary. How is the activity and expression of DacA, the diadenylate cyclase, regulated? Our transcript analyses indicate that *dacA-ybbR* transcripts peak during the RB phase of growth, similar to most genes in *Chlamydia*. One possibility not easily tested due to the obligate intracellular nature of *Chlamydia* is that c-di-AMP production depletes ATP levels, which then reduce $\sigma^{66}$ activity as described above. Our data indicate YbbR is necessary to activate DacA function since expressing membrane-localized or ΔTM isoforms of DacA alone only modestly increased c-di-AMP levels (*Figure 1E*). However, overexpressing DacA alone did negatively impact chlamydial growth, suggesting that the balance between DacA and YbbR levels is important, with YbbR being the limiting factor for c-di-AMP production. For example, too much DacA insertion into the membrane without binding to YbbR may disrupt the membrane biology of the RB. This is supported by the fact that we observed no detrimental effects of overexpressing ΔTM isoforms of either wild-type or mutant (D164N) DacA. Indeed, overexpressing ΔTMDacA(D164N) did not alter c-di-AMP levels. Therefore, YbbR itself may be a target for regulation. As a monotopic transmembrane protein, YbbR may be degraded by proteases such as the inner membrane-associated FtsH or the periplasmic proteases HtrA or Tsp. This would presumably shut down c-di-AMP synthesis. Alternatively, DacA may interact with other membrane proteins, and overexpressing it alone may impair the function of this binding partner(s), resulting in the observed phenotypes. However, this possibility contradicts the effects of overexpressing the D164N isoform, which blocked c-di-AMP production. These experiments were conducted in the presence of the

chromosomally expressed copy of DacA, and we suggest that the mutant isoform acts as a dominant negative by binding and/or interfering with the wild-type DacA or by titrating away YbbR from the wild-type DacA. This is also supported by the fact that overexpressing the ΔTMDacA(D164N) isoform had no impact on c-di-AMP levels or chlamydial growth. In contrast, knocking down *dacA-ybbR* transcripts will reduce, but not completely eliminate, DacA and YbbR proteins, which may result in residual c-di-AMP levels as compared to overexpressing the DacA mutant (either alone or with YbbR in *dacA-opMut*) as well as the phenotypic differences between these strains. Further work is needed to test these possibilities. However, our data are clear in linking increased c-di-AMP levels to chlamydial developmental progression.

Even though diverse functions of c-di-AMP in other bacteria have been reported previously, this is the first time c-di-AMP has been described as a checkpoint in chlamydial development. It is possible that the levels of c-di-AMP act as a de facto means for monitoring the bacterial population within the inclusion or for their overall developmental status. Canonical bacteria can use quorum sensing to detect bacteria at the population level and to communicate with other species (*Miller and Bassler, 2001*). However, *C. trachomatis* lacks homologues of genes related to quorum sensing (*Stephens et al., 1998*). In addition, secondary differentiation occurs asynchronously, and this feature is different

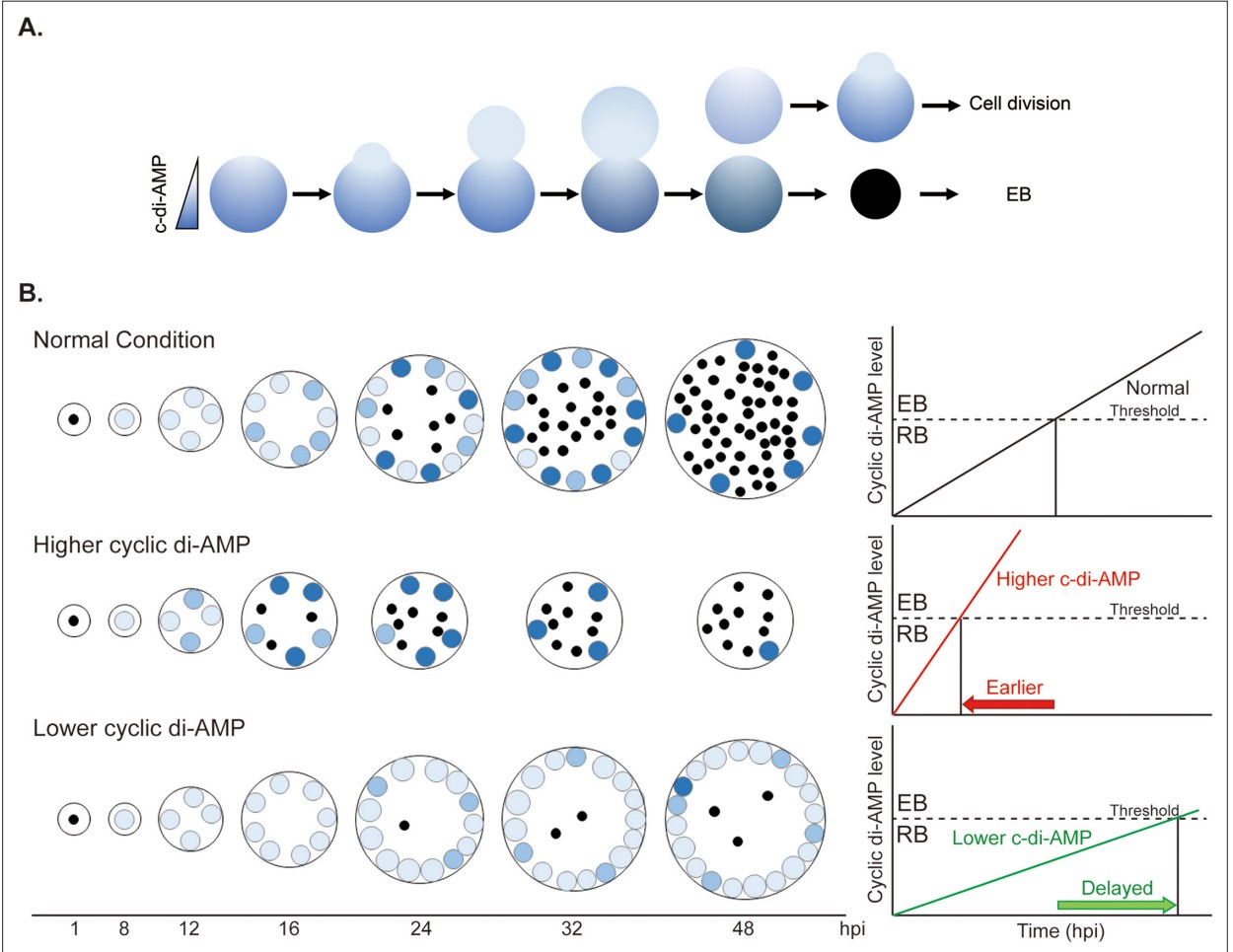

**Figure 8.** A working model illustrating how c-di-AMP impacts the onset of secondary differentiation in *C. trachomatis*. (**A**) During reticulate body (RB) division, there is a gradient of c-di-AMP that forms between the mother and daughter cell. The lighter blue represents lower levels of c-di-AMP, the darker blue represents higher levels of c-di-AMP. In the depicted scenario, the daughter cell is able to divide again, having not reached a critical threshold of c-di-AMP. In contrast, the mother cell accumulates sufficient c-di-AMP to trigger secondary differentiation to an EB. (**B**) A model illustrating how c-di-AMP levels impact secondary differentiation progression. Shown is the 'normal' condition of wild-type bacteria, bacteria overexpressing DacA and YbbR (higher cyclic di-AMP), and bacteria expressing the CRISPRi system targeting the *dacA* promoter (lower cyclic di-AMP). Depending on the amount of c-di-AMP produced, secondary differentiation can either be triggered earlier or later as shown. The black dots represent EBs; the bigger and blue-colored circles show the RB cells. The brightness of blue reflects the c-di-AMP level.

from a quorum sensing mechanism. Our model accounts for the asynchronicity of secondary differentiation. As RBs accumulate c-di-AMP and divide, it is likely the c-di-AMP will be distributed unevenly between the mother and daughter cells since *Chlamydia* divides through an asymmetric budding mechanism (*Abdelrahman et al., 2016*; *Figure 8*). This will lead to two cells with different levels of c-di-AMP, one of which may then accumulate enough c-di-AMP to trigger secondary differentiation while the other continues to divide (*Figure 8*). This is consistent with a recent model proposing that a population of division-competent RBs is maintained throughout the developmental cycle (*Chiarelli et al., 2023*). Similar uneven distribution of post-translational factors between asymmetrically dividing mother and daughter cells would also impact other recently described signals for differentiation, including the levels of ClpC protein or overall redox via AhpC antioxidant levels (*Jensen et al., 2025*; *Singh and Ouellette, 2025*).

## Materials and methods
### Organisms, cell lines, and cell culture
Wild-type (ATCC, Manassas, VA) and STING KO (Dr. Frank van Kuppeveld; Utrecht University) HeLa, human cervical epithelial-derived, and McCoy (kind gift of Dr. Harlan Caldwell, NIH), mouse fibroblast-derived, cell lines were cultured at 37 °C with 5% $CO_2$ in Dulbecco's Modified Eagle Medium (DMEM; Gibco, Waltham, MA, #10-569-044) containing 10% fetal bovine serum (FBS; Hyclone, Logan, UT, #SH30396.03) and 10 μg/mL gentamicin (Gibco, Waltham, MA, #15710072). Cell lines were verified by STR profiling (ATCC) and monitoring of cell morphology. *C. trachomatis* serovar L2 (434/Bu) lacking the endogenous plasmid (-pL2; kind gift of Dr. Ian Clarke, Univ. Southampton) was used for transformation. All cell cultures and chlamydial stocks were routinely tested for Mycoplasma contamination using the Mycoplasma PCR detection kit and confirmed negative (Sigma, St. Louis, MO, #MP0035-1KT). For *E. coli*, NEB10β competent cells (New England Biolabs, Ipswich, MA, #C3019H) were used for the amplification of pBOMB-derivative vectors. *E. coli* was grown at 30 °C in LB media. All chemicals and antibiotics were obtained from Sigma unless otherwise noted.

### Cloning
The list of the vectors and primers used in this study is detailed in *Supplementary file 3*. Target genes were amplified by PCR with Phusion DNA polymerase (NEB, #M0530L) using 10 ng *C. trachomatis* L2 genomic DNA or appropriate vectors as a template. Some DNA segments were directly synthesized as a gBlock fragment (Integrated DNA Technologies, Coralville, IA). If plasmids were used as a template, then we treated the PCR product with DpnI enzyme to remove templates. The PCR products were purified using a PCR purification kit (Qiagen, Hilden, Germany, #28506). The HiFi Assembly reaction master mix (NEB, #E2621X) was used following the manufacturer's manual in conjunction with plasmids pBOMB (*Ouellette et al., 2021*) linearized with EagI and KpnI or pBOMBL12CRia (empty vector) linearized with BamHI. The linearized plasmids were also dephosphorylated with FastAP (ThermoFisher, #EF0652). The products of the HiFi reaction were transformed into NEB10β competent cells (NEB) and plated on LB agar with appropriate antibiotics. Plasmids were subsequently isolated using a mini-prep kit (Qiagen, #27106) and verified by plasmid digest and sequencing from individual colonies grown overnight in LB broth with appropriate antibiotic selection.

### Transformation of *Chlamydia trachomatis*
McCoy cells were plated in a six-well plate the day before beginning the transformation procedure. *C. trachomatis* serovar L2 without plasmid (-pL2) resuspended with Tris-CaCl$_2$ buffer (10 mM Tris-Cl pH 7.5, 50 mM $CaCl_2$) was incubated with 2 μg plasmid at room temperature for 30 min. During this step, McCoy cells were washed with 2 mL Hank's Balanced Salt Solution (HBSS) media containing $Ca^{2+}$ and $Mg^{2+}$ (Corning, Corning, NY #21–023-CV). After that, McCoy cells were infected with the transformants in 2 mL HBSS per well. The plate was centrifuged at 400 × g for 15 min at room temperature and incubated at 37 °C for 15 min. The inoculum was aspirated, and 2 mL 1 X DMEM containing 10% FBS and 10 μg/mL gentamicin was added per well. At 8 hr post- infection (hpi), 1 μg/mL cycloheximide and either 1 or 2 U/mL penicillin G or 500 μg/mL spectinomycin were added, and the plate was incubated at 37 °C until 48 hpi. At 48 hpi, the transformants were harvested and infected onto a new

McCoy cell monolayer. These harvest and infection steps were repeated every 44–48 hpi until fluorescent, antibiotic-resistant inclusions were observed.

## (RT-)qPCR

HeLa cells were infected with chlamydial transformants at an MOI of 0.5. At 10 hpi, 5 nM anhydrotetracycline (aTc) was added or not to the culture medium. Total RNA and DNA were harvested at this time point from duplicate wells not treated with aTc. At 14 and 24 hpi, total RNA and DNA were collected using Trizol (Invitrogen, #15596018) and DNeasy Tissue (Qiagen, #69506) kit, respectively, as described elsewhere (*Hatch and Ouellette, 2023*). After DNase (Invitrogen, #69506) treatment of total RNA, cDNA was synthesized using Superscript III reverse transcriptase (Invitrogen, #18-080-085). After diluting the cDNA 10-fold, 5 µl of the diluted cDNA was used as a template for qPCR. Equal masses of genomic DNA were used from each of the samples to quantify chlamydial genomes, which were used to normalize transcript data as described (*Ouellette et al., 2005*). For both cDNA and gDNA samples, triplicate qPCR reactions were prepared using 2 X SYBR Green (Applied Biosystems, #A25778) in a total volume of 25 µL per well. Standard cycling conditions were used with a melting curve analysis to verify products. Transcripts and genome copies were assessed from at least three biological replicates.

## Indirect immunofluorescence assay (IFA)

HeLa cells were infected with chlamydial transformants as above. At 10 hpi, 5 nM aTc was added or not, and the infected cells were fixed with fixing solution (3.2% formaldehyde and 0.022% glutaraldehyde in 1 X DPBS) for 2 min and permeabilized with 90% MeOH for 1 min at 10.5 hpi or 24 hpi. The fixed cells were labeled with primary antibodies, including rabbit anti-DacA (custom anti-peptide antibody targeting the C-terminal TRNERKTNPIISWMRKK prepared by Pacific Immunology, Ramona, CA), goat anti-major outer-membrane protein (MOMP; Meridian, Memphis, TN), and mouse or rabbit anti-six histidine tag (Genscript, Piscataway, NJ, #A00186-100 and Abcam, Cambridge, UK, #AB213204, respectively). To visualize the primary antibodies, donkey anti-goat (488) (Invitrogen, #A32814) donkey anti-mouse (405) (Invitrogen, #A48257), or donkey anti-rabbit antibodies (594) (Invitrogen, #A21207) were used as secondary antibodies. Coverslips were observed using a Zeiss AxioImager.Z2 with Apotome2 as noted in the figure legends.

## IFU measurement

HeLa cells were infected with chlamydial transformants as above. At 10 hpi, 5 nM aTc was added or not to the culture medium. At the indicated times, infected cells were harvested in 1 mL 2SP media then frozen at –80 °C. After thawing the lysates, the samples were serially 1:10 diluted and used to infect HeLa cells seeded in 24-well plates. At 24 hpi, the number of GFP- expressing inclusions was counted from 30 fields of view to calculate the IFUs from the original sample. Three biological replicates were performed.

## Cyclic di-AMP measurement

HeLa cells were infected with the indicated chlamydial transformants as above. At 10 hpi, expression of the constructs was induced or not with 5 nM aTc, and, at 16 and 24 hpi, samples were prepared for measuring the level of cyclic di-AMP. After aspirating the media, the infected cell monolayers were washed with 1 X PBS and resuspended with B-Per Bacterial Cell Lysis Buffer (Pierce, Appleton, WI, #PI90078). Samples were vortexed for 1 min and then centrifuged at 13,300×g for 15 min at 4°C. The levels of cyclic di-AMP from the supernatants of the cell lysates were quantified using the Cyclic di-AMP ELISA kit (Cayman, Ann Arbor, MI, #501960-STRIP) following the manufacturer's instructions (*Underwood et al., 2014*).

## Preparation of RNA sequencing samples

Samples were prepared as reported previously (*Hatch and Ouellette, 2023*). Briefly, the transformants of *dacA*op and *dacA*-KD were infected into HeLa cells. At 10 hpi, the constructs were induced or not with 5 nM aTc. RNA samples were prepared from *dacA*op-infected cells and *dacA*-KD-infected cells at 16 hpi and 24 hpi, respectively. 20 µg RNA samples were treated with DNase to remove DNA contamination using DNA-free Turbo kit (Invitrogen, #AM1907) according to the manufacturer's

instructions. Ribosomal RNA was depleted from samples using the MICROBEnrich (Thermo, #AM1901) and MICROBExpress kits (Thermo, AM1905) following the manufacturer's instructions. RNA samples were processed for sequencing by the UNMC Genomics Core Facility. The resultant libraries from the individual samples were multiplexed and subjected to 100 bp paired-read sequencing to generate approximately 60 million pairs of reads per sample on an Illumina NovaSeq 6000 sequencer in the UNMC Genomics Core facility. The original fastq format reads were trimmed by fqtirm tool (https://ccb.jhu.edu/software/fqtrim) to remove adapters, terminal unknown bases (Ns) and low quality 3' regions (Phred score <30). The trimmed fastq files were processed by FastQC (*Andrews, 2010*) for quality control. *Chlamydia trachomatis* 434/Bu bacterial reference genome and annotation files were downloaded from Ensembl (http://bacteria.ensembl.org/Chlamydia_trachomatis_434_bu/Info/Index). Sequencing data were analyzed by the Bioinformatics and Systems Biology Core (BSBC). The trimmed fastq files were mapped to *Chlamydia trachomatis* 434/Bu by CLC Genomics Workbench 23 for RNAseq analyses.

## Statistical analysis

To analyze the statistical significance between uninduced and induced samples of qPCR and IFU data, we used two-sample equal variance Student's t-test. For the levels of cyclic di-AMP data, we used two-sample equal variance Student's t-test.

## Acknowledgements

This study was supported in part by an NIH/NIGMS award (R35GM124798) and in part by an NIH/NIAID award (R21AI180574) to SPO and by start-up funds from UNMC. The UNMC Genomics Core Facility receives partial support from the National Institute for General Medical Science (NIGMS) INBRE - P20GM103427-19, as well as the National Cancer Institute, the Fred & Pamela Buffett Cancer Center Support Grant- P30CA036727. This publication's contents are the sole responsibility of the authors and do not necessarily represent the official views of the NIH or NIGMS. The authors would like to thank Dr. Frank van Kuppeveld (Utrecht University) for STING-KO HeLa cells, Dr. Harlan Caldwell (NIAID/NIH) for McCoy cells, and Dr. Ian Clarke (University of Southampton) for the plasmidless *C. trachomatis* serovar L2 strain. In addition, we thank the UNMC Genomics Core Facility and the Bioinformatics and Systems Biology Core Facility for RNA sequencing and analysis services, respectively.

# Additional information

## Funding

| Funder | Grant reference number | Author |
| --- | --- | --- |
| National Institutes of Health | 1R35GM124798 | Scot P Ouellette |
| National Institutes of Health | 1R21AI180574 | Scot P Ouellette |

The funders had no role in study design, data collection and interpretation, or the decision to submit the work for publication.

## Author contributions

Junghoon Lee, Conceptualization, Formal analysis, Validation, Investigation, Visualization, Methodology, Writing – original draft; Scot P Ouellette, Conceptualization, Resources, Formal analysis, Supervision, Funding acquisition, Visualization, Project administration, Writing – review and editing

## Author ORCIDs

Junghoon Lee ⓘ https://orcid.org/0000-0002-1948-3087
Scot P Ouellette ⓘ https://orcid.org/0000-0002-3721-6839

Reviewer #2 (Public review): https://doi.org/10.7554/eLife.104240.4.sa1

Author response https://doi.org/10.7554/eLife.104240.4.sa2

## Additional files

### Supplementary files
MDAR checklist

Supplementary file 1. Generalized representation of sequencing efficiency.

Supplementary file 2. RNA sequencing results.

Supplementary file 3. List of plasmids and primers used in the study.

### Data availability
The raw and processed RNA sequencing reads in fastq format have been deposited in the Gene Expression Omnibus (GEO; http://www.ncbi.nlm.nih.gov/geo/) under accession no. GSE252732. All other data are available in the main text or the supplementary materials.

The following dataset was generated:

| Author(s) | Year | Dataset title | Dataset URL | Database and Identifier |
|---|---|---|---|---|
| Lee J, Ouellette SP | 2025 | Cyclic di-AMP drives secondary differentiation in *Chlamydia trachomatis* | https://www.ncbi.nlm.nih.gov/geo/query/acc.cgi?acc=GSE252732 | NCBI Gene Expression Omnibus, GSE252732 |

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
