## [Editor Report · eLife Assessment]

In this **useful** study, ectopic expression and knockdown strategies were used to assess the effects of increasing and decreasing Cyclic di-AMP on the developmental cycle in Chlamydia. The authors **convincingly** demonstrate that overexpression of the dacA-ybbR operon results in increased production of c-di-AMP and early expression of the transitionary gene hctA and late gene omcB. Whilst the authors have attempted to revise the submission, the model proposed in the revised manuscript is still not fully supported by the data presented.

---

## [Referee Report · Reviewer #2 (Public review)]

This manuscript describes the role of the production of c-di-AMP on the chlamydial developmental cycle. The main findings remain the same. The authors show that overexpression of the dacA-ybbR operon results in increased production of c-di-AMP and early expression of transitionary and late genes. The authors also knocked down the expression of the dacA-ybbR operon and reported a modest reduction in the expression of both hctA and omcB. The authors conclude with a model suggesting the amount of c-di-AMP determines the fate of the RB, continued replication, or EB conversion.

Overall, this is a very intriguing study with important implications however the data is very preliminary and the model is very rudimentary. The data support the observation that dramatically increased c-di-AMP has an impact on transitionary gene expression and late gene expression suggesting dysregulation of the developmental cycle. This effect goes away with modest changes in c-di-AMP (detaTM-DacA vs detaTM-DacA (D164N)). However, the model predicts that low levels of c-di-AMP delays EB production is not not well supported by the data. If this prediction were true then the growth rate would increase with c-di-AMP reduction and the data does not show this. The levels of of c-di-AMP at the lower levels need to be better validated as it seems like only very high levels make a difference for dysregulated late gene expression. However, on the low end it's not clear what levels are needed to have an effect as only DacAopMut and DacAopKD show any effects on the cycle and the c-di-AMP levels are only different at 24 hours.

The data still do not support the overall model.

In Figure 1 the authors show at 24 hpi.

DacA overexpression increases cdiAMP to ~4000 pg/ml

DacAmut overexpression reduces cdiAMP dramatically to ~256 pg/ml

DacATM overexpression increases cdiAMP to ~4000 pg/ml.

DacAmutTM overexpression does not seem to change cdiAMP ~1500 pg/ml .

dacAKD decreases cdiAMP to ~300 pg/ml .

dacAKDcom increased cdiAMP to ~8000 pg/ml.

DacA-ybbRop overexpression increased cdiAMP to ~500,000 pg/ml.

DacA-ybbRopmut ~300 pg/ml.

However in Figure 2 the data show that overexpression of DacA (cdiAMP ~4000 pg/ml) did not have a different phenotype than over expression of the mutant (cdiAMP ~256 pg/ml). HctA expression down, omcB expression down, euo not much change, replication down, and IFUs down. Additionally, Figure 3 shows no differences in anything measured although cdiAMP levels were again dramatically different. DacATM overexpression (~4000 pg/ml) and DacAmutTM (~1500). This makes it unclear what cdiAMP is doing to the developmental cycle.

In Figure 4 the authors knockdown dacA (dacA-KD) and complement the knockdown (dacA-KDcom) dacAKD decreases cdiAMP (~300) while DacA-KDcom increases cdiAMP much above wt (~8000).

KD decreased hctA and omcB at 24hpi. Complementation resulted in a moderate increase in hctA at a single time point but not at 24 hpi and had no effect on euo or omcB expression. Importantly, complementation decreased the growth rate. Based on the proposed model, growth rate should increase as the chlamydia should all be RBs and replicating and not exiting the cell cycle to become EBs (not replicating). Interestingly reducing cdiAMP levels by over expressing DacAmut (~256 pg/ml) did not have an effect on the cycle but the reduction in cdiAMP by knockdown of dacA (~300 pg/ml) did have a moderate effect on the cycle.

For Figure 5 DacA-ybbRop was overexpressed and this increased cdiAMP dramatically ~500,000 pg/ml as compared to wt ~1500. This increased hctA only at an early timepoint and not at 24hpi and again had no effect on omcB or euo. Overexpression of the operon with the mutation DacA-ybbRopmut reduced cdiAMP to ~300 pg/ml and this showed a reduction in growth rate similar to dacAmut but a more dramatic decrease in IFUs.

Overall:

DacA overexpression increases cdiAMP to ~4000 pg/ml (decreased everything except euo)

DacAmut overexpression reduces cdiAMP dramatically (~256 pg/ml). (decreased everything except euo)

DacATM overexpression increases cdiAMP to ~4000 pg/ml (no changes noted)

DacAmutTM overexpression does not seem to change cdiAMP ~1500 pg/ml (no changes noted)

dacAKD decrease cdiAMP to ~300 pg/ml (decreased everything except euo)

dacAKDcom increased cdiAMP to ~8000 pg/ml (decreases growth rate, increase hctA a little but not omcB)

DacA-ybbRop overexpression increased cdiAMP to ~500,000 pg/ml (decreases growth rate, increase hctA a little but not omcB)

DacA-ybbRopmut ~300 pg/ml (decreased everything except euo)

Overall, the data show that increasing cdiAMP only has a phenotype if it is dramatically increased, no effect at 4000 pg/ml. Decreasing cdiAMP has a consistent effect, decreased growth rate, IFU, hctA expression and omcB expression. However, if their proposed model was correct and low levels of cdiAMP blocked EB conversion then more chlamydial cells would be RBs (dividing cells) and the growth rate should increase. Conversely, if cdiAMP levels were dramatically raised then all RBs would all convert and the growth rate would be very low. When cdiAMP was raised to ~4000 pg/ml there was no effect on the growth rate. However, an increase to ~8000 pg/ml resulted in a significant decrease but growth continued. Increasing cdAMP to ~500,000 pg/ml had less of an impact on the growth rate. Overall, the data does not cleanly support the proposed model.

---

## [Author Response]

The following is the authors’ response to the current reviews

**Reviewer #2 (Public review):**
This manuscript describes the role of the production of c-di-AMP on the chlamydial developmental cycle. The main findings remain the same. The authors show that overexpression of the dacA-ybbR operon results in increased production of c-di-AMP and early expression of transitionary and late genes. The authors also knocked down the expression of the dacA-ybbR operon and reported a modest reduction in the expression of both hctA and omcB. The authors conclude with a model suggesting the amount of c-di-AMP determines the fate of the RB, continued replication, or EB conversion.Overall, this is a very intriguing study with important implications however the data is very preliminary and the model is very rudimentary. The data support the observation that dramatically increased c-di-AMP has an impact on transitionary gene expression and late gene expression suggesting dysregulation of the developmental cycle. This effect goes away with modest changes in c-di-AMP (detaTM-DacA vs detaTM-DacA (D164N)). However, the model predicts that low levels of c-di-AMP delays EB production is not not well supported by the data. If this prediction were true then the growth rate would increase with c-di-AMP reduction and the data does not show this. The levels of of c-di-AMP at the lower levels need to be better validated as it seems like only very high levels make a difference for dysregulated late gene expression. However, on the low end it's not clear what levels are needed to have an effect as only DacAopMut and DacAopKD show any effects on the cycle and the c-di-AMP levels are only different at 24 hours.

These appear to be the same comments the reviewer presented last time, so we will reiterate our prior points here and elsewhere. We do not think and nor do we predict that low c-di-AMP levels should increase growth rate (as measured by gDNA levels), and this conclusion cannot be drawn from our data. Rather, we predict that the inability to accumulate c-di-AMP should delay production of EBs, and this is what the data show. The reviewer has applied their own subjective (and erroneous) interpretation to the model. The asynchronicity of the normal developmental cycle means RBs continue to replicate as EBs are forming, so gDNA levels cannot be used as the sole metric for determining RB levels. We show that reduced c-di-AMP levels reduce EB levels as well as transcripts associated with late stages of development. The parsimonious interpretation of these data support that low c-di-AMP levels delay progression through the developmental cycle consistent with our model.

The data still do not support the overall model.

We disagree. We have presented quantified data that include appropriate controls and statistical tests, and the reviewer has not disputed that or pointed to additional experiments that need to be performed. The reviewer has imposed a subjective interpretation of our model based on their own biases. A reader is free, of course, to disagree with our model, but a reviewer should not block a manuscript based on such a disagreement if no experimental flaws have been identified.

In Figure 1 the authors show at 24 hpi.

We also showed data from 16hpi, which is a more relevant timepoint for assessing premature transition to EBs. In contrast, the 24hpi is more important for assessing developmental effects of reduced c-di-AMP levels.

DacA overexpression increases cdiAMP to ~4000 pg/mlDacAmut overexpression reduces cdiAMP dramatically to ~256 pg/mlDacATM overexpression increases cdiAMP to ~4000 pg/ml.DacAmutTM overexpression does not seem to change cdiAMP ~1500 pg/ml .dacAKD decreases cdiAMP to ~300 pg/ml .dacAKDcom increased cdiAMP to ~8000 pg/ml.DacA-ybbRop overexpression increased cdiAMP to ~500,000 pg/ml.DacA-ybbRopmut ~300 pg/ml.However in Figure 2 the data show that overexpression of DacA (cdiAMP ~4000 pg/ml) did not have a different phenotype than over expression of the mutant (cdiAMP ~256 pg/ml). HctA expression down, omcB expression down, euo not much change, replication down, and IFUs down. Additionally, Figure 3 shows no differences in anything measured although cdiAMP levels were again dramatically different. DacATM overexpression (~4000 pg/ml) and DacAmutTM (~1500). This makes it unclear what cdiAMP is doing to the developmental cycle.

As we have explained in the text and in response to reviewer comments on previous rounds of review, overexpressing the full-length WT or mutant DacA is detrimental to developmental cycle progression for reasons that have nothing to do with c-di-AMP levels (likely disrupting membrane function), since, as the reviewer notes, the WT DacA deltaTM strain had similar c-di-AMP levels but no negative effects on growth/development. If we had not presented the effects of overexpressing the individual isoforms, then a reviewer would surely have requested such, which is why we present these data even though they don’t seem to support our model. This is an honest representation of our findings. The reviewer seems intent on nitpicking a minor datapoint that seems to contradict the rest of the manuscript while ignoring or not carefully reading the rest of the manuscript.

In Figure 4 the authors knockdown dacA (dacA-KD) and complement the knockdown (dacA-KDcom)dacAKD decreases cdiAMP (~300) while DacA-KDcom increases cdiAMP much above wt (~8000).KD decreased hctA and omcB at 24hpi. Complementation resulted in a moderate increase in hctA at a single time point but not at 24 hpi and had no effect on euo or omcB expression.

By 24hpi, late gene transcripts are being maximally produced during a normal developmental cycle. It is unclear why the reviewer thinks that these transcripts should be elevated above this level in any of our strains that prematurely transition to EBs. There is no basis in the literature to support such an assumption. As we noted in the text, the dacA-KDcom strain phenocopied the dacAop OE strain, and we showed RNAseq data and EB production curves for the latter that support our conclusions of the effect of increased c-di-AMP levels on developmental progression.

Importantly, complementation decreased the growth rate.

Yes, since the c-di-AMP levels breached the “EB threshold” at 16hpi, it causes premature transition to EBs, which do not replicate their gDNA, at an earlier stage of the cycle when fewer organisms are present. Therefore, the gDNA levels are decreased at 24hpi, which is consistent with our model.

Based on the proposed model, growth rate should increase as the chlamydia should all be RBs and replicating and not exiting the cell cycle to become EBs (not replicating).

This is a spurious conclusion from the reviewer. As we clearly showed, the dacA-KDcom did not restore a wild-type phenotype and instead mimicked the dacAop OE strain. This was commented on in the text.

Interestingly reducing cdiAMP levels by over expressing DacAmut (~256 pg/ml) did not have an effect on the cycle but the reduction in cdiAMP by knockdown of dacA (~300 pg/ml) did have a moderate effect on the cycle.

This is again a spurious conclusion from the reviewer. The dacAMut and dacA-KD strains are distinct. As noted in the text and above for DacA WT OE, overexpressing the DacAMut similarly disrupts organism morphology, which is different from dacA-KD. These strains should not be directly compared because of this. This point has been previously highlighted in the text (in Results and Discussion).

For Figure 5 DacA-ybbRop was overexpressed and this increased cdiAMP dramatically ~500,000 pg/ml as compared to wt ~1500. This increased hctA only at an early timepoint and not at 24hpi and again had no effect on omcB or euo.

As we explained in prior reviews, our RNAseq data more comprehensively assessed transcripts for the dacAop OE strain. These data show convincingly that late gene transcripts (not just hctA and omcB) are elevated earlier in the developmental cycle. Again, it is not clear why the reviewer should expect that late gene transcripts should be higher in these strains than they are during a normal developmental cycle. This is not part of our model and appears to be a bias that the reviewer has imposed that is not supported by the literature.

Overexpression of the operon with the mutation DacA-ybbRopmut reduced cdiAMP to ~300 pg/ml and this showed a reduction in growth rate similar to dacAmut but a more dramatic decrease in IFUs.

As we described in the text, in earlier revisions, and above, the dacAMut OE strain has distinct effects unrelated to c-di-AMP levels and, therefore, should not be compared to other strains in terms of linking its c-di-AMP levels to its phenotype.

Overall:DacA overexpression increases cdiAMP to ~4000 pg/ml (decreased everything except euo)DacAmut overexpression reduces cdiAMP dramatically (~256 pg/ml). (decreased everything except euo)DacATM overexpression increases cdiAMP to ~4000 pg/ml (no changes noted)DacAmutTM overexpression does not seem to change cdiAMP ~1500 pg/ml (no changes noted)dacAKD decrease cdiAMP to ~300 pg/ml (decreased everything except euo)dacAKDcom increased cdiAMP to ~8000 pg/ml (decreases growth rate, increase hctA a little but not omcB)DacA-ybbRop overexpression increased cdiAMP to ~500,000 pg/ml (decreases growth rate, increase hctA a little but not omcB)DacA-ybbRopmut ~300 pg/ml (decreased everything except euo)Overall, the data show that increasing cdiAMP only has a phenotype if it is dramatically increased, no effect at 4000 pg/ml.

Yes, this clearly shows there is a threshold - as we hypothesize! However, these thresholds are more important at the 16hpi timepoint not 24hpi (which the reviewer is referencing) when assessing premature transition to EBs. We specifically highlighted in our prior revision in Figure 1E this EB threshold to make this point clearer for the reader. Once the threshold is breached, then the overall c-di-AMP levels become irrelevant as the RBs have begun their transition to EBs.

Decreasing cdiAMP has a consistent effect, decreased growth rate, IFU, hctA expression and omcB expression. However, if their proposed model was correct and low levels of cdiAMP blocked EB conversion then more chlamydial cells would be RBs (dividing cells) and the growth rate should increase.

The only effect should be normal gDNA levels, which is what we see in the dacA-KD. Given the asynchronicity of a normal developmental cycle in which RBs continue to replicate as EBs are still forming, there is no basis to assume gDNA levels should increase under these conditions for the dacA-KD strain at 24hpi.

Conversely, if cdiAMP levels were dramatically raised then all RBs would all convert and the growth rate would be very low.

We agree. This is what is reflected by the dacAop OE and dacA-KDcom strains, with reduced gDNA levels at 24hpi since organisms have transitioned to EBs at an earlier time post-infection.

When cdiAMP was raised to ~4000 pg/ml there was no effect on the growth rate.

Yes, because it had not breached the EB threshold at 16hpi – consistent with our model! The reviewer is confusing effects of elevated c-di-AMP at 24hpi when they should be assessed at the 16hpi timepoint for strains overproducing this molecule.

However, an increase to ~8000 pg/ml resulted in a significant decrease but growth continued.

If the reviewer is referring to the dacA-KDcom strain, then this is not accurate. gDNA levels were decreased in this strain at 24hpi when the c-di-AMP levels were increased compared to the WT (mCherry OE) control at 16hpi, indicating this strain had breached the “EB threshold” and initiated conversion to EBs at an earlier timepoint post-infection when fewer organisms were present.

Increasing cdAMP to ~500,000 pg/ml had less of an impact on the growth rate.

It is not clear what this conclusion is based on and what the reviewer is comparing to. This is a subjective assessment not based on our data.

Overall, the data does not cleanly support the proposed model.

It is an unfortunate aspect of biology, particularly for obligate intracellular bacteria – a challenging experimental system on which to work, that the data are not always “clean”. The overall effects of increased c-di-AMP levels on chlamydial developmental cycle progression we have documented support our model, and we think the reader, as always, should make their own assessment.

The following is the authors’ response to the original reviews.

**Reviewer #2 (Public review):**
This manuscript describes the role of the production of c-di-AMP on the chlamydial developmental cycle. The main findings remain the same. The authors show that overexpression of the dacA-ybbR operon results in increased production of c-di-AMP and early expression of transitionary and late genes. The authors also knocked down the expression of the dacA-ybbR operon and reported a modest reduction in the expression of both hctA and omcB. The authors conclude with a model suggesting the amount of c-di-AMP determines the fate of the RB, continued replication, or EB conversion.Overall, this is a very intriguing study with important implications however, the data is very preliminary, and the model is very rudimentary. The data support the observation that dramatically increased c-di-AMP has an impact on transitionary gene expression and late gene expression suggesting dysregulation of the developmental cycle. This effect goes away with modest changes in c-di-AMP (detaTM-DacA vs detaTM-DacA (D164N)). However, the model predicts that low levels of c-di-AMP delays EB production is not not well supported by the data. If this prediction were true then the growth rate would increase with c-di-AMP reduction and the data does not show this.

Thank you for the comments. We have apparently not adequately communicated our predictions and the model. We do not think and nor do we predict that low c-di-AMP levels should increase growth rate, and there is no basis in any of our data to support that. Rather, we predict that the inability to accumulate c-di-AMP should delay production of EBs, and this is what the data show. We have clarified this in the text (line 89 paragraph).

The levels of c-di-AMP at the lower levels need to be better validated as it seems like only very high levels make a difference for dysregulated late gene expression. However, on the low end it's not clear what levels are needed to have an effect as only DacAopMut and DacAopKD show any effects on the cycle and the c-di-AMP levels are only different at 24 hours.

Our hypothesis is that increasing concentrations of c-di-AMP within a given RB is a signal for it to undergo secondary differentiation to the EB, and the data support this as noted by the reviewers. Again, we stress that low levels of c-di-AMP are irrelevant to the model. We have revised Figure 1E to indicate the level of c-di-AMP in the control strain at the 24hpi timepoint that coincides with increased EB levels. We hope this will further clarify the goals of our study. That a given strain might be below the EB control is not relevant to the model beyond indicating that it has not reached the necessary threshold for triggering secondary differentiation.

The authors responded to reviewers' critiques by adding the overexpression of DacA without the transmembrane region. This addition does not really help their case. They show that detaTM-DacA and detaTM-DacA (D164N) had the same effects on c-di-AMP levels but the figure shows no effects on the developmental cycle.

As it relates directly to the reviewer’s point, the delta-TM strains did not show the same level of c-di-AMP. It may be that the reviewer misread the graph. The purpose of testing these strains was to show that the negative effects of overexpressing full-length WT DacA were due to its membrane localization. Both the FL and deltaTM-DacA (WT) overexpression had equivalent c-di-AMP levels even though the delta-TM overexpression looked like the mCherry-expressing strain based on the measured parameters. This shows that the c-di-AMP levels were irrelevant to the phenotypes observed when overexpressing these WT isoforms. For the mutant isoforms, the delta-TM looked like the mCherry-expressing control while the FL isoform was negatively impacted for reasons we described in the Discussion (e.g., dominant negative effect). In addition, at 16hpi, neither delta-TM strain had c-di-AMP levels that approached the 24h control as denoted in Figure 1E (dashed line) and in the text, which explains why these strains did not show increased late gene transcripts at an earlier timepoint like the dacAop and dacA-KDcom strains.

Describing the significance of the findings:The findings are important and point to very exciting new avenues to explore the important questions in chlamydial cell form development. The authors present a model that is not quantified and does not match the data well.

We respectfully disagree with this assessment as noted above in response to the reviewer’s critique. All of our data are quantified and support the hypothesis as stated.

Describing the strength of evidence:The evidence presented is incomplete. The authors do a nice job of showing that overexpression of the dacA-ybbR operon increases c-di-AMP and that knockdown or overexpression of the catalytically dead DacA protein decreases the c-di-AMP levels. However, the effects on the developmental cycle and how they fit the proposed model are less well supported.Overall this is a very intriguing finding that will require more gene expression data, phenotypic characterization of cell forms, and better quantitative models to fully interpret these findings.

It is not clear what quantitative models the reviewer would prefer, but, ultimately, it is up to the reader to decide whether they agree or not with the model we present. The data are the data, and we have tried to present them as clearly as possible. We would emphasize that, with the number of strains we have analyzed, we have presented a huge amount of data for a study with an obligate intracellular bacterium. As a comparison, most publications on Chlamydia might use a handful of transformant strains, if any. Given the cost and time associated with performing such studies, it is prohibitive to attempt all the time points that one might like to do, and it is not clear to us that further studies will add to or alter the conclusions of the current manuscript.

**Reviewer #2 (Recommendations for the authors):**
Minor critiquesThe graphs have red and blue lines but the figure legends are red and black. It would be better if these matched.

Changed.

For Figure 1C. The labels are not very helpful. It's not clear what is HeLa vs mCherry. I believe it is uninfected vs Chlamydia infected.

Changed.